# A Universal Analysis of Large-Scale Regularized Least Squares Solutions

**Ashkan Panahi**
Department of Electrical and Computer Engineering
North Carolina State University
Raleigh, NC 27606
apanahi@ncsu.edu

**Babak Hassibi**
Department of Electrical Engineering
California Institute of Technology
Pasadena, CA 91125
hassibi@caltech.edu

## Abstract

A problem that has been of recent interest in statistical inference, machine learning and signal processing is that of understanding the asymptotic behavior of regularized least squares solutions under random measurement matrices (or dictionaries). The Least Absolute Shrinkage and Selection Operator (LASSO or least-squares with $\ell_1$ regularization) is perhaps one of the most interesting examples. Precise expressions for the asymptotic performance of LASSO have been obtained for a number of different cases, in particular when the elements of the dictionary matrix are sampled independently from a Gaussian distribution. It has also been empirically observed that the resulting expressions remain valid when the entries of the dictionary matrix are independently sampled from certain non-Gaussian distributions. In this paper, we confirm these observations theoretically when the distribution is sub-Gaussian. We further generalize the previous expressions for a broader family of regularization functions and under milder conditions on the underlying random, possibly non-Gaussian, dictionary matrix. In particular, we establish the universality of the asymptotic statistics (e.g., the average quadratic risk) of LASSO with non-Gaussian dictionaries.

## 1 Introduction

During the last few decades, retrieving structured data from an incomplete set of linear observations has received enormous attention in a wide range of applications. This problem is especially interesting when the ambient dimension of the data is very large, so that it cannot be directly observed and manipulated. One of the main approaches is to solve regularized least-squares optimization problems that are tied to the underlying data model. This can be generally expressed as:

$$\min_{\mathbf{x}} \frac{1}{2}\|\mathbf{y} - \mathbf{A}\mathbf{x}\|_2^2 + f(\mathbf{x}), \tag{1}$$

where $\mathbf{x} \in \mathbb{R}^m$ and $\mathbf{y} \in \mathbb{R}^n$ are the desired data and observation vectors, respectively. The matrix $\mathbf{A} \in \mathbb{R}^{m \times n}$ is the sensing matrix, representing the observation process, and the regularization function $f : \mathbb{R}^n \to \mathbb{R}$ imposes the desired structure on the observed data. When $f$ is convex, the optimization in (1) can be solved reliably with a reasonable amount of calculations. In particular, the case where $f$ is the $\ell_1$ norm is known as the LASSO, which has been extremely successful in retrieving sparse data vectors. During the past years, random sensing matrices $\mathbf{A}$ have been widely used and studied in the context of the convex regularized least squares problems. From the perspective of data retrieval, this choice is supported by a number of studies in the so-called Compressed Sensing (CS) literature, which show that under reasonable assumptions, random matrices may lead to good performance [1, 2, 3].

An interesting topic, addressed in the recent compressed sensing literature (and also considered here), is to understand the behavior of the regularized least squares solution in the asymptotic case, where $m$ and $n$ grow to infinity with a constant ratio $\gamma = m/n$. For this purpose, a scenario is widely considered, where $\mathbf{y}$ is generated by the following linear model:

$$\mathbf{y} = \mathbf{A}\mathbf{x}_0 + \boldsymbol{\nu}, \tag{2}$$

where $\mathbf{x}_0$ is the true structured vector and $\boldsymbol{\nu}$ is the noise vector, here assumed to consist of independent centered Gaussian entries, with equal variances $\sigma^2$. Then, it is desired to characterize the statistical behavior of the optimal solution $\hat{\mathbf{x}}$ of (1), also called the estimate, and the error $\mathbf{w} = \hat{\mathbf{x}} - \mathbf{x}_0$. More specifically, we are interested in the asymptotic empirical distribution[1] of the estimate $\hat{\mathbf{x}}$ and the error $\mathbf{w}$, when the sensing matrix is also randomly generated with independent and identically distributed entries. Familiar examples of such matrices are Gaussian and Bernoulli matrices.

## 1.1 Previous Work

Analyzing linear least squares problems with random matrices has a long history. The behavior of the unregularized solution, or that of ridge regression (i.e., $\ell_2-$regularization) is characterized by the singular values of $\mathbf{A}$ which is well-understood in random matrix theory [4, 5]. However, a general study of regularized solutions became prominent with the advent of compressed sensing, where considerable effort has been directed toward an analysis in the sense explained above. In compressed sensing, early works focused on the LASSO ($\ell_1$ regularization), sparse vectors $\mathbf{x}_0$ and the case, where $\boldsymbol{\nu} = 0$ [6]. These works aimed at providing conditions to guarantee perfect recovery, meaning $\mathbf{w} = 0$, and established the Restricted Isometry Property (RIP) as a deterministic perfect recovery condition. This condition is generally difficult to verify [7, 8]. It was immediately observed that under mild conditions, random matrices satisfy the RIP condition with high probability, when the dimensions grow to infinity with a proper ratio $\gamma$ [9, 10]. Soon after, it was discovered that the RIP condition was unnecessary to undertake the analysis for random matrices. In [11], an "RIP-less" theory of perfect recovery was introduced.

Despite some earlier attempts [12, 13], a successful error analysis of the LASSO for Gaussian matrices was not obtained until the important paper [14], where it was shown by the analysis of so-called approximate message passing (AMP) that for any pseudo Lipschitz function $\psi; \mathbb{R}^2 \to \mathbb{R}$, and defining $\hat{x}_i, x_{i0}$ as the $i^{\text{th}}$ elements of $\hat{\mathbf{x}}$ and $\mathbf{x}_0$, respectively, the sample risk

$$\frac{1}{n}\sum_{k=1}^{n} \psi(\hat{x}_i, x_{i0})$$

converges to a value that can be precisely computed. As a special case, the asymptotic value of the scaled $\ell_2$ norm of the error $\mathbf{w}$ is calculated by taking $\psi(\hat{x}_i, x_{i0}) = (\hat{x}_i - x_{i0})^2$. In [15], similar results are obtained for M-estimators using AMP. Fundamental bounds for linear estimation with Gaussian matrices are also recently provided in [16]. Another remarkable direction of progress was made in a series of papers, revolving around an approach, first developed by Gordon in [17], and introduced to the compressed sensing literature by Stojnic in [18]. Employing Gordon's approach, [19] provides the analysis of a broad range of convex regularized least squares problems for Gaussian sensing matrices. Exact expressions are provided in this work only for asymptotically small noise. In [20] this result is utilized to provide the exact analysis of the LASSO for general noise variance, confirming the earlier results in [14]. Some further investigations are recently provided in [21] and [22].

When there is no measurement noise, universal (non-Gaussian) results on the phase transition for the number of measurements that allows perfect recovery of the signal have been recently obtained in [23]. Another special case of ridge ($\ell_2$) regression is studied in [24]. The technical approach in [23] is different from ours. Furthermore, the current paper considers measurement noise and is concerned with the performance of the algorithm and not on the phase transitions for perfect recovery. In [25], the so-called Lindeberg approach is proposed to study the asymptotic behavior of the LASSO. This is similar in spirit to our approach. However, the study in [25] is more limited than ours, in the sense that it only establishes universality of the expected value of the optimal cost when the LASSO is restricted to an arbitrary rectangular ball. Some stronger bounds on the error risk of LASSO are

established in [23, 26], which are sharp for asymptotically small noise or large $m$. However, to the best of our knowledge, there have not been any exact universal results of the generality as ours in the literature. It is also noteworthy that our results can be predicted by the replica symmetry (RS) method as partially developed in [27]. Another recent area where the connection of RS and performance of estimators has been rigorously established is low rank matrix estimation [28, 29]

## 2   Main Results

Our contributions are twofold: First, we generalize the expressions in [21] and [20] for a more general case of arbitrary separable regularization functions $f(\mathbf{x})$ where with an abuse of notation

$$f(\mathbf{x}) = \sum_{i=1}^{n} f(x_i) \tag{3}$$

and the function $f$ on the right hand side is a real function $f(x) : \mathbb{R} \to \mathbb{R}$. Second, we show that our result is universal, which precisely means that our expressions are independent of the distribution (law) of the i.i.d sensing matrix.

In general, we tie the asymptotic behavior of the optimization in (1) to the following two-dimensional optimization, which we refer to as the *essential* optimization:

$$C_f(\gamma, \sigma) = \max_{\beta \geq 0} \min_{p > 0} \left\{ \frac{p\beta(\gamma-1)}{2} + \frac{\gamma\sigma^2\beta}{2p} - \frac{\gamma\beta^2}{2} + \mathbb{E}\left[ S_f\left( \frac{\beta}{p}, p\Gamma + X \right) \right] \right\}, \tag{4}$$

where $X$ and $\Gamma$ are two independent random variables, distributed by an arbitrary distribution $\xi$ and standard Gaussian p.d.f, respectively. Further, $S_f(.,.)$ denotes the proximal function of $f$, which is defined by

$$S_f(q, y) = \min_x \frac{q}{2}(x - y)^2 + f(x). \tag{5}$$

with the minimum located at $\hat{x}_f(q, y)$. If the solution $\left( \hat{p} = \hat{p}(\gamma, \sigma), \hat{\beta} = \hat{\beta}(\gamma, \sigma) \right)$ of (4) is unique, then we define the random variables

$$\hat{X} = \hat{X}_{f, \xi, \sigma, \gamma} = \hat{x}_f\left( \frac{\hat{\beta}}{\hat{p}}, \hat{p}\Gamma + X \right), \quad W = \hat{X} - X$$

Our result can be expressed by the following theorem:

**Theorem 1** *Suppose that the entries of* $\mathbf{A}$ *are first generated independently by a proper distribution[2]* $\mu$ *and next scaled by* $1/\sqrt{m}$. *Moreover, assume that the true vector* $\mathbf{x}_0$ *is randomly generated and has i.i.d. entries with some distribution* $\xi$. *Then,*

- *The optimal cost in* (1)*, scaled by* $\frac{1}{n}$*, converges in probability to* $C_f(\gamma, \sigma)$,

- *The empirical distributions of the solution* $\hat{\mathbf{x}}$ *and the error* $\mathbf{w}$ *weakly converge to the distribution of* $\hat{X}$ *and* $W$*, respectively,*

*if one of the following holds:*

1. *The real function* $f$ *is strongly convex.*

2. *The real function* $f$ *equals* $\lambda|x|$ *for some* $\lambda > 0$, $\mu$ *is further* $\sigma_s^2$*-sub-Gaussian[3] and the "effective sparsity"* $M_0 = \Pr(\hat{X} \neq 0)$ *is smaller than a constant depending on* $\mu, \lambda, \gamma, \sigma$. *For example,* $M_0 \leq \rho/2$ *works where*

$$\rho \log 9 + H(\rho) \leq \min \left\{ 1, \frac{9}{8\sigma_s^2} + \frac{1}{2} \log \frac{8\sigma_s^2}{9} \right\}$$

*and* $H(\rho) = -\rho \log \rho - (1-\rho) \log(1-\rho)$ *is the binary entropy function[4].*

We include more detailed and general results, as well as the proofs in the supplementary material. In the rest of this paper, we discuss the consequences of Theorem 1, especially for the case of the LASSO, and give a sketch of the proof of our results.

## 3 Remarks and Numerical Results

In this section, we discuss few issues arising from our analysis.

### 3.1 Evaluation of Asymptotic Values

A crucial question related to Theorem 1 and the essential optimization is how to calculate the optimal parameters in (4). Here, our purpose is to provide a simple instruction for solving the optimization in (4). Notice that (4) is a min-max optimization over the pair $(p, \beta)$ of real positive numbers. We observe that there exists an appealing structure in this optimization, which substantially simplifies its numerical solution:

**Theorem 2** *For any fixed $\beta > 0$, the objective function in* (4) *is convex over $p$. For any fixed $\beta$, denote the optimal value of the inner optimization (over $p$) of* (4) *by $\psi(\beta)$. Then, $\psi$ is a concave function of $\beta$.*

Using Theorem 2, we may reduce the problem of solving (4) into a sequence of single dimensional convex optimization problems (line searches). We assume a derivative-free[5] algorithm $\texttt{alg}(\phi)$, such as dichotomous search (See the supplement for more details), which receives as an input (an oracle of) a convex function $\phi$ and returns its optimal value and its optimal point over $[0 \infty)$. Denote the cost function of (4) by $\phi(p, \beta)$. This means that $\psi(\beta) = \min_p \phi(p, \beta)$. If $\phi(p, \beta)$ is easy to calculate, we observe that $\texttt{alg}(\phi(p, \beta))$ for a fixed $\beta$ is an oracle of $\psi(\beta)$. Since $\psi(\beta)$ is now easy to calculate we may execute $\texttt{alg}(\psi(\beta))$ to obtain the optimal parameters.

#### 3.1.1 Derivation for LASSO

To apply the above technique, we require a fast method to evaluate the objective function in (4). Here, we provide the expressions for the case of LASSO with $f(x) = \lambda|x|$, which is originally formulated in [30]. For this case, we assume that the entries of the true vector $\mathbf{x}_0$ are non-zero and standard Gaussian with probability $0 \leq \kappa \leq 1$. In other words, $\xi = \kappa \mathcal{N} + (1 - \kappa)\delta_0$, where $\mathcal{N}$ and $\delta_0$ are standard Gaussian and the Dirac measures on $\mathbb{R}$, respectively. Then, we have that

$$\mathbb{E}\left[S_f\left(\frac{\beta}{p}, p\Gamma + X\right)\right] = \kappa\sqrt{1 + p^2}F(\frac{\beta}{p}\sqrt{1 + p^2}) + (1 - \kappa)pF(\beta)$$

where

$$F(q) = \frac{\lambda e^{-\frac{\lambda^2}{2q^2}}}{2\sqrt{2\pi}} - \frac{q}{2}\left(1 + \frac{\lambda^2}{q^2}\right)Q(\frac{\lambda}{q}) + \frac{q}{4}$$

The function $Q(.)$ is the Gaussian tail Q-function. We may replace the above expression in the definition of essential optimization to obtain $\hat{p}, \hat{\beta}$ and the random variables $\hat{X}, W$ in Section 2. Now, let us calculate $\|\mathbf{w}\|_2^2/n$ by taking expectation over empirical distribution of $\mathbf{w}$. Using Theorem 1, we obtain the following term for the asymptotic value of $\|\mathbf{w}\|_2^2/n$:

$$\mathbb{E}(W^2) = \kappa J\left(\frac{\lambda p}{\beta}, p, 1\right) + (1 - \kappa)J\left(\frac{\lambda p}{\beta}, p, 0\right)$$

where

$$J(\epsilon, p, \alpha) = \alpha^2 + 2\left(p^2 + \epsilon^2 - \alpha^2\right)Q\left(\frac{\epsilon}{\sqrt{\alpha^2 + p^2}}\right) - 2\epsilon\sqrt{\frac{\alpha^2 + p^2}{2\pi}}\exp\left(-\frac{\epsilon^2}{2(\alpha^2 + p^2)}\right)$$

Figure 1a depicts the average value $\|\mathbf{w}\|_2^2/n$ over 50 independent realizations of the LASSO, including independent Gaussian sensing matrices with $\gamma = 0.5$, sparse true vectors with $\kappa = 0.2$ and Gaussian

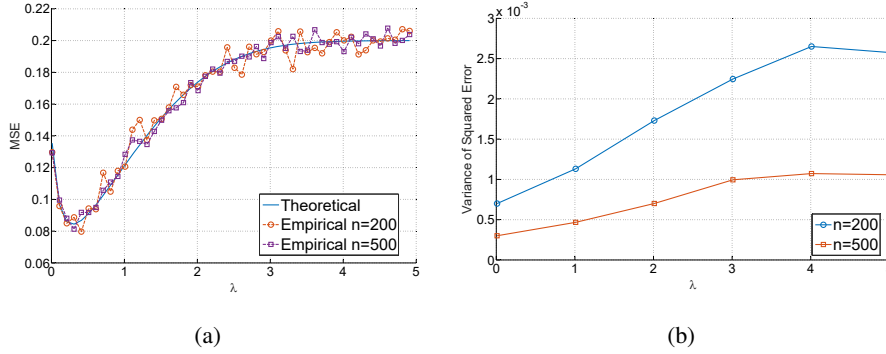

|     |     |
| --- | --- |
| (a) | (b) |

Figure 1: a)The sample mean of the quadratic risk for different values of $\lambda$, compared to its theoretical value. The average is taken over 50 trials. b) The sample variance of the quadratic risk for different values of $\lambda$. The average is taken over 1000 trials.

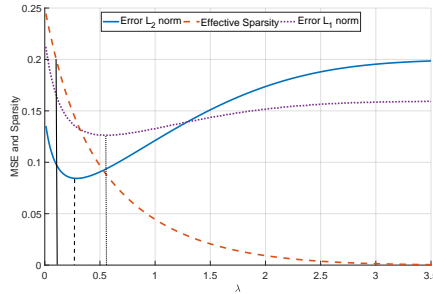

Figure 2: Asymptotic error $\ell_1$ and squared $\ell_2$ norms, as well as the solution sparsity. Their corresponding optimal $\lambda$ values are depicted by vertical lines.

noise realizations with $\sigma^2 = 0.1$. We consider two different problem sizes $n = 200, 500$. As seen, the sample mean, which approximates the statistical mean $\mathbb{E}(\|\mathbf{w}\|_2^2/n)$, agrees with the theoretical results above.

Figure 1b examines the convergence of the error 2-norm by depicting the sample variance of $\|\mathbf{w}\|_2^2/n$ for the two cases above with $n = 200, 500$. Each data point is obtained by 1000 independent realizations. As seen, the case $n = 500$ has a smaller variance, which indicates that as dimensions grow the variance of the quadratic risk vanishes and it converges in probability to its mean. Another interesting phenomenon in Figure 1b is that the larger values of $\lambda$ are associated with larger uncertainty (variance), especially for smaller problem sizes.

The asymptotic analysis allows us to decide an optimal value of the regularization parameter $\lambda$. Figure 2 shows few possibilities. It depicts the theoretical values for the error squared $\ell_2$ and $\ell_1$ norms as well as the sparsity of the solution. The (effective[6]) sparsity can be calculated as

$$M_0 = \Pr(\hat{X} \neq 0) = 2(1-\kappa)Q\left(\frac{\lambda}{\beta}\right) + 2\kappa Q\left(\frac{\lambda p}{\beta\sqrt{1+p^2}}\right)$$

The expression for the $\ell_1$ norm can be calculated similar to the $\ell_2$ norm, but does not have closed form and is calculated by a Monte Carlo method. We observe that at the minimal error, both in $\ell_2$ and $\ell_1$ senses, the solution is sparser than the true vector ($\kappa = 0.2$). On the contrary, adjusting the sparsity to the true one slightly increases the error $\ell_2$ norm. As expected, the sparsity of the solution decreases monotonically with increasing $\lambda$.

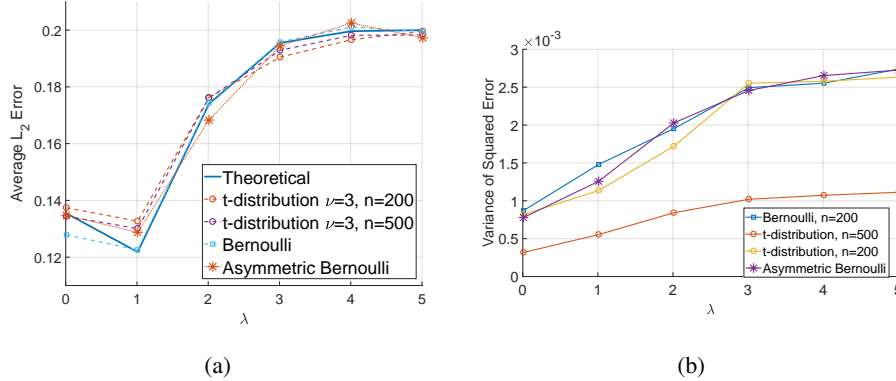

Figure 3: (a) The average LASSO error $\ell_2$ norm (b) the sample variance of LASSO error $\ell_2$ norm for different matrices.

## 3.2 Universality and Heavy-tailed Distributions

In the previous section, we demonstrated numerical results, which were generated by Gaussian matrices. Here, we focus on universality. In Section 2, our results are under some regularity assumptions for the sensing matrix. For the LASSO, we require sub-Gaussian entries and low sparsity, which is equivalent to a large regularization value. Here, we examine these conditions in three cases: First, a centered Bernoulli matrix where each entry $-1$ or $1$ with probability $1/2$. Second, a matrix distributed by Student's t-distribution with 3 degrees of freedom $\nu = 3$ and scaled to possess unit variance. Third, an asymmetric Bernoulli matrix where each entry is either $3$ or $-1/3$ with probabilities $0.1$ and $0.9$, respectively. Figure 3 shows the error $\ell_2$ norm and its variance for the LASSO case. As seen, all cases follow the predicted asymptotic result. However, the results for the t-matrix and the asymmetric distribution is beyond our analysis, since t-distribution does not possess finite statistical moments of an order larger than 2 and the asymmetric case possesses non-vanishing third moment. This indicates that our universal results hold beyond the limit assumed in this paper. However, we are not able to prove it with our current technique.

## 3.3 Remarks on More General Universality Results

As we explain in the supplementary document, Theorem 1 is specialized from a more general result. Here, we briefly discuss the main aspects of our general result. When the regularization is not separable ($f$ is non-separable) our analysis may still guarantee universality of its behavior. However, we are not able to evaluate the asymptotic values anymore. Instead, we relate the behavior of a general sensing matrix to a reference choice, e.g. a Gaussian matrix. For example, we are able to show that if the optimal objective value in (1) converges to a particular value for the reference matrix, then it converges to exactly the same value for other suitable matrices in Theorem 1. The asymptotic optimal value may remain unknown to us. The universality of the optimal value holds for a much broader family of regularizations than the separable ones in (3). For example, "weakly separable" functions of the form

$$f(\mathbf{x}) = \frac{\sum\limits_{i,j} f(x_i, x_j)}{n}$$

or the generalized fused LASSO [31, 32] are simply seen to possess universal optimal values. One important property of our generalized result is that if we are able to establish optimal value universality for a particular regularization function $f(\mathbf{x})$, then we automatically have a similar result for $f(\boldsymbol{\Psi}\mathbf{x})$, where $\boldsymbol{\Psi}$ is a fixed matrix satisfying certain regularity conditions[7]. This connects our analysis to the analysis of generalized LASSO [33, 34]. Moreover, substituting $f(\boldsymbol{\Psi}\mathbf{x})$ in (1) and changing the optimization variable to $\mathbf{x}' = \boldsymbol{\Psi}\mathbf{x}$, we obtain (1) where $\mathbf{A}$ is replaced by $\mathbf{A}\boldsymbol{\Psi}^{-1}$. Hence, our approach enables us to obtain further results on sensing matrices of the form $\mathbf{A}\boldsymbol{\Psi}^{-1}$, where $\mathbf{A}$ is i.i.d and $\boldsymbol{\Psi}$ is deterministic. We postpone more careful analysis to future papers.

It is worth mentioning that we obtain Theorem 1 about the separable functions in the light of the same principle: We simply connect the behavior of the error for an arbitrary matrix to the Gaussian one. In this particular case, we are able to carry out the calculations over Gaussian matrices with the well-known techniques, developed for example in [18] and briefly explained below.

## 4  Technical Discussion

### 4.1  An Overview of Approach

In this section, we present a crude sketch of our mathematical analysis. Our aim is to show the main ideas without being involved in mathematical subtleties. There are four main elements in our analysis, which we address in the following.

#### 4.1.1  From Optimal Cost to the Characteristics of the Optimal Point

In essence, we study the optimal values of optimizations such as the one in (1). Studying the optimal solution directly is much more difficult. Hence, we employ an indirect method, where we connect an arbitrary real-valued characteristic (function) $g$ of the optimal point to the optimal value of a set of related optimizations. This is possible through the following simple observation:

**Lemma 1** *We are to minimize a convex function $\phi(\mathbf{x})$ on a convex domain $D$ and suppose that $\mathbf{x}^*$ is a minimal solution. Further, $g(x)$ is such that the function $\phi + \epsilon g$ remains convex when $\epsilon$ is in a symmetric interval $[-e\ e]$. Define $\Phi(\epsilon)$ as the minimal value of $\phi + \epsilon g$ on $D$. Then, $\Phi(\epsilon)$ is concave on $[-e\ e]$ and $g(\mathbf{x}^*)$ is its subgradient at $\epsilon = 0$.*

As a result of Lemma 1, the increments $\Phi(\epsilon) - \Phi(0)$ and $\Phi(0) - \Phi(-\epsilon)$ for positive values of $\epsilon$ provide lower and upper bounds for $g(\mathbf{x}^*)$, respectively. We use these bounds to prove convergence in probability. However, Lemma 1 requires $\phi + \epsilon g$ to remain convex for both positive and negative values of $\epsilon$. It is now simple to see that choosing a strongly convex function for $f$ and a convex function with bounded second derivative for $g$ ensures this requirement.

#### 4.1.2  Lindeberg's Approach

With the above approach, we only need to focus on the universality of the optimal values. To obtain universal results, we adopt the method that Lindeberg famously developed to obtain a strong version of the central limit theorem [35]. Lindeberg's approach requires a reference case, where asymptotic properties are simple to deduce. Then, similar results are proved for an arbitrary case by considering a finite chain of intermediate problems, starting from the reference case and ending at the desired case. In each step, we are able to analyze the change in the optimal value and show that the sum of these changes cannot be substantial for asymptotically large cases. In our study, we take the optimization in (1) with a Gaussian matrix $\mathbf{A}$ as reference. In each step of the chain, we replace one Gaussian row of $\mathbf{A}$ with another one with the target distribution. After $m$ step, we arrive at the desired case. At each step, we analyze the change in the optimal value by Taylor expansion, which shows that the change is of second order and is $o(\frac{1}{m})$ (in fact $O(\frac{1}{m^{5/4}})$) with high probability, such that the total change is bounded by $o(1)$. For this, we require strong convexity and bounded third derivatives. This shows universality of the optimal value.

#### 4.1.3  Asymptotic Results For Gaussian Matrices

Since we take Gaussian matrices as reference in the Lindeberg's approach, we require a different machinery to analyze the Gaussian case. The analysis of (1) for the Gaussian matrices is considered in [19]. Here, we briefly review this approach and specialize it in some particular cases. Let us start by defining the following so-called Key optimization, associated with (1):

$$\phi_n(\mathbf{g}, \mathbf{x}_0) = \max_{\beta > 0} \min_{\mathbf{v} \in \mathbb{R}^n} \frac{m\beta}{n} \sqrt{\sigma^2 + \frac{\|\mathbf{v}\|_2^2}{m}} + \beta \frac{\mathbf{g}^T \mathbf{v}}{n} - \frac{m}{2n}\beta^2 + \frac{f_n(\mathbf{v} + \mathbf{x}_0)}{n} \qquad (6)$$

where $\mathbf{g}$ is a $n-$dimensional standard Normal random vector, independent of other variables. Then, [19] shows that in case $\mathbf{A}$ is generated by a standard Gaussian random variable and $\phi_n(\mathbf{g}, \mathbf{x}_0)$ converges in probability to a value $C$, then the optimal value in (1) also converges to $C$. The

consequences of this observation are thoroughly discussed in [20]. Here, we focus on a case, where $f(\mathbf{x})$ is separable as in 3. For this case, The Key optimization in (6) can be simplified and stated as in the following theorem (See [30]).

**Theorem 3** *Suppose that $\mathbf{A}$ is generated by a Gaussian distribution, $\mathbf{x}_0$ is i.i.d. with distribution $\xi$ and $f(\mathbf{x})$ is separable as in (1). Furthermore, $m/n \rightarrow \gamma \in \mathbb{R}_{\geq 0}$. Then, the optimal value of the optimization in (1), converges in probability to $C_f(\gamma, \sigma)$ defined in Section 2.*

Now, we may put the above steps together to obtain the desired result for the strongly convex functions: Lindeberg's approach shows that the optimal cost is universal. On the other hand, the optimal cost for Gaussian matrices is given by $C_f(\gamma, \sigma)$. We conclude that $C_f(\gamma, \sigma)$ is the universal limit of the optimal cost. Now, we may use the argument in Lemma 1 to obtain a characteristic $g$ at the optimal point. For this, we may take for example regularizations of the form $f + \epsilon g$, which by the previous discussion converges to $C_{f+\epsilon g}$. Then, $g(\hat{\mathbf{x}})$ becomes equal to $\mathrm{d}C_{f+\epsilon g}/\mathrm{d}\epsilon$ at $\epsilon = 0$, which by further calculations leads to the result in Theorem 1 [8].

### 4.1.4   Final Step: The LASSO

The above argument fails for the LASSO with $f(x) = \lambda|x|$ because it lacks strong convexity. Our remedy is to start from an "augmented approximation" of the LASSO with $f(x) = \lambda|x| + \epsilon x^2/2$ and to show that the solution of the approximation is stable in the sense that removing the term $\epsilon x^2/2$ does not substantially change the optimal point. We employ a slightly modified argument in [12], which requires two assumptions: a) The solution is sparse. b) The matrix $\mathbf{A}$ is sufficiently restricted-isometric. The condition on restricted isometry is satisfied by assuming sub-Gaussian distributions [36], while the sparsity of the solution is given by $M_0$. The assumption that $M_0$ is sufficiently small allows the argument in [12] to hold in our case, which ensures that the LASSO solution remains close to the solution of the augmented LASSO and the claims of Theorem 1 can be established for the LASSO. However, we are able to show that the optimal value of the LASSO is close to that of the augmented LASSO without any requirement for sparsity. This can be found in the supplementary material.

## 5   Conclusion

The main purpose of this study was to extend the existing results about the convex regularized least squares problems in two different directions, namely more general regularization functions and non-Gaussian sensing matrices. In the first direction, we tied the asymptotic properties for general separable convex regularization functions to a two-dimensional optimization that we called the essential optimization. We also provided a simple way to calculate asymptotic characteristics of the solution from the essential optimization. In the second direction, we showed that the asymptotic behavior of regularization functions with certain regularity conditions is independent of the distribution (law) of the sensing matrix. We presented few numerical experiments which validated our results. However, these experiments suggest that the universality of the asymptotic behavior holds beyond our assumptions.

### 5.1   Future Research

After establishing the convergence results, a natural further question is the rate of convergence. The properties of regularized least squares solutions with finite size is not well-studied even for the Gaussian matrices. Another interesting subject for future research is to consider random sensing matrices, which are not necessarily identically distributed. We believe that our technique can be generalized to a case with independent rows or columns instead of elements. A similar generalization can be obtained by considering true vectors with a different structure. Moreover, we introduced a number of cases such as generalized LASSO [34] and generalized fused Lasso [32], where our analysis shows universality, but the asymptotic performance cannot be calculated. Calculating the asymptotic values of these problems for a reference choice, such as Gaussian matrices is an interesting subject of future study.

## Footnotes

[1]Empirical distribution of a vector $\mathbf{x}$ is a measure $\nu$ where $\nu(A)$ is the fraction of entries in $\mathbf{x}$ valued in $A$.

[2]Here, a proper distribution is the one with vanishing first, third and fifth moments, unit variance and finite fourth and sixth moments.

[3]A centered random variable $Z$ is $\sigma_s^2$-sub-Gaussian if $\mathbb{E}(e^{rZ}) \leq e^{\frac{\sigma_s^2 r^2}{2}}$ holds for every $r \in \mathbb{R}$.

[4]In this paper, all logarithms are to natural base ($e$).

[5]It is also possible to use the derivative-based algorithms, but it requires to calculate the derivatives of $S_f$ and $\psi$. We do not study this case.

[6]Since we establish weak convergence of the empirical distribution, this number does not necessarily reflect the number of exactly zero elements in $\hat{\mathbf{x}}$, but rather the "infinitesimally" small ones.

[7]More precisely, we require $\boldsymbol{\Psi}$ to have a strictly positive smallest singular value and a bounded third operator norm.

[8]The expression for $g(\mathbf{w})$ is found in a similar way, but requires some mathematical preparations, which we express later.

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
