[Supplementary Material · supplement.pdf]

# Supplement: A Universal Analysis of Large-Scale Regularized Least Squares Solutions

**Ashkan Panahi**
Department of Electrical and Computer Engineering
North Carolina State University
Raleigh, NC 27606
apanahi@ncsu.edu

**Babak Hassibi**
Department of Electrical Engineering
California Institute of Technology
Pasadena, CA 91125
hassibi@caltech.edu

## 1 Main Results in Detail

Here, we express the results of the theorem in more details. We introduce complementary results and change some expressions for convenience in the mathematical development. Especially, we introduce a new set of more detailed results, which are connected to the results of the paper as we shortly explain.

The main feature of our analysis is that it is universal. This precisely means that our expressions are independent of the distribution (law) of the i.i.d sensing matrix as long as it belongs to the following family of scaled-regular matrices:

**Definition 1.** *We call a random matrix $\mathbf{A}$ regular if it consists of independent and identical distributed entries with vanishing first, third and fifth moments, unit variance and finite fourth and sixth moments. We call a $m \times n$ random matrix $\mathbf{A}$ scaled-regular if it can be written as $\mathbf{A} = \mathbf{A}'/\sqrt{m}$, where $\mathbf{A}'$ is regular.*

We also present the analysis in terms of a general real-valued characteristic of the optimal solution $\hat{\mathbf{x}}$ or the error $\mathbf{w}$. This characteristic is defined by a *characteristic function* $g : \mathbb{R}^n \to \mathbb{R}$. Accordingly, we are interested in calculating asymptotic values for $g(\mathbf{w})$ and $g(\hat{\mathbf{x}})$, which we prove to be independent of the law of the matrix $\mathbf{A}$ under mild conditions. We are able to calculate these values only when the functions are separable, i.e. there exist (with an abuse of notation) real functions $f(x), g(x) : \mathbb{R} \to \mathbb{R}$ such that

$$f(\mathbf{x}) = \sum_{i=1}^{n} f(x_i), \quad g(\mathbf{x}) = \sum_{i=1}^{n} g(x_i). \tag{1}$$

Finally, we assume that the true vector $\mathbf{x}_0$ is randomly generated and has i.i.d. entries with some distribution $\xi$. Notice that $\xi$ represents the structure in $\mathbf{x}_0$. For example, sparse vectors can be generated by a distribution $\xi$ that contains an atom at 0, i.e. $\xi(\{0\}) > 0$.

Since we study the asymptotic behavior of the regularized least squares, we technically consider a family of problems as in (1 in Paper) with a growing size $n$. Hence, we may use subscript $n$ to clarify the relation with size. For example the functions $f, g$ are written as $f_n, g_n$, respectively. We also denote the optimal value in (1 in Pape) by $\Phi_n = \Phi_n(\mathbf{A}, \boldsymbol{\nu}, \mathbf{x}_0)$ and use the notations $\hat{\mathbf{x}} = \hat{\mathbf{x}}(\mathbf{A}, \boldsymbol{\nu}, \mathbf{x}_0)$ and $\mathbf{w} = \mathbf{w}(\mathbf{A}, \boldsymbol{\nu}, \mathbf{x}_0)$ to emphasize the dependence of the estimate and the error on the realizations of $\mathbf{A}, \boldsymbol{\nu}, \mathbf{x}_0$.

We split our results into two groups: strongly convex regularizations and the original LASSO ($\ell_1$ regularization). This is because the $\ell_1$ norm is not strongly convex and its analysis requires a different treatment. Both results are based on the notion of *essential optimization*, which we explain first.

## 1.1 Essential Optimization

For a case with separable functions $f, g$ and independent $\xi-$distributed $\mathbf{x}_0$, we observe that the asymptotic behavior of the regularized least squares problems is reflected by the following two-dimensional optimization that we call the essential optimization:

$$C_f(\gamma, \sigma) = \max_{\beta \geqslant 0} \min_{p > 0} \left\{ \frac{p\beta(\gamma - 1)}{2} + \frac{\gamma\sigma^2\beta}{2p} - \frac{\gamma\beta^2}{2} + \mathcal{E}\left[ S_f\left( \frac{\beta}{p}, p\Gamma + X \right) \right] \right\}, \tag{2}$$

where $X$ and $\Gamma$ are two independent random variables, distributed by $\xi$ and standard Gaussian p.d.f, respectively. Further, $S_f(.\,,.)$ denotes the proximity function of $f$, which is defined by

$$S_f(q, y) = \min_x \frac{q}{2}(x - y)^2 + f(x). \tag{3}$$

with the minimum located at $\hat{x}(q, y)$. If the solution $\left( \hat{p} = \hat{p}(\gamma, \sigma), \hat{\beta} = \hat{\beta}(\gamma, \sigma) \right)$ of (2) is unique, then we define

$$L_{f,g}(\gamma, \sigma) = \mathcal{E}\left( g\left( \hat{x}_f\left( \frac{\hat{\beta}}{\hat{p}}, \hat{p}\Gamma + X \right) - X \right) \right), \quad M_{f,g}(\gamma, \sigma) = \mathcal{E}\left( g\left( \hat{x}_f\left( \frac{\hat{\beta}}{\hat{p}}, \hat{p}\Gamma + X \right) \right) \right). \tag{4}$$

## 1.2 Strongly Convex Regularization

Our analysis assumes both differentiable and non-differentiable regularization functions $f$. For differentiable functions, we consider strongly convex ones with absolutely bounded third derivative. Referring to these functions as smooth-regular, we consider for non-differentiable functions the ones that are obtained as a *uniform* limit of smooth-regular functions. We call them regular functions. In other words, the set of regular regularization functions is the uniform closure of all strongly convex functions with absolutely bounded third derivative. For simplicity, we only report the result for smooth-regular functions here and postpone the more general case to Section 1.4.

For the characteristic function $g$, we simply take the set of all convex functions with bounded second and third derivatives. We do not consider the non-differentiable functions. Also, notice that once we establish convergence results for these characteristic functions we may take any affine combination of a finite number of them, which extends the result to a large family of non-convex functions. In particular, we can establish the universality result for the characteristic function $g(x) = \chi_{[x_0\ \infty)}(x)$, which is 1, if $x \in [x_0\ \infty)$, and 0, otherwise. This choice corresponds to the empirical distribution $F_{x_0}(.)$ that counts the number of entries larger than a particular value $x_0$ in its argument.

**Theorem 1.** *Suppose that $f$ is smooth-regular and $g$ is a convex function with bounded second and third derivatives. Assume that $\mathbf{A}$ is a scaled-regular random matrix, $\boldsymbol{\nu}$ is a centered i.i.d Gaussian vector with variance $\sigma^2$ and $\mathbf{x}_0$ is i.i.d with distribution $\xi$, such that $\mathcal{E}((f'(X))^2)$ is finite for a $\xi-$distributed random variable $X$. Moreover $n, m$ grow, such that $m/n \to \gamma$. Then,*

    *1. We have that*

$$\Phi_n(\mathbf{A}, \boldsymbol{\nu}, \mathbf{x}_0) \to_p C_f(\gamma, \sigma) \tag{5}$$

    *2. If the solutions of (2) $\hat{p} = \hat{p}(\gamma, \sigma), \hat{\beta} = \hat{\beta}(\gamma, \sigma)$ is unique, then*

$$\frac{g_n(\mathbf{w}(\mathbf{A}, \boldsymbol{\nu}, \mathbf{x}_0))}{n} \to_p L_{f,g}(\gamma, \sigma), \quad \frac{g_n(\hat{\mathbf{x}}(\mathbf{A}, \boldsymbol{\nu}, \mathbf{x}_0))}{n} \to_p M_{f,g}(\gamma, \sigma) \tag{6}$$

    *3. For every $x \in \mathbb{R}$, we have that*

$$\frac{F_x(\mathbf{w}(\mathbf{A}, \boldsymbol{\nu}, \mathbf{x}_0))}{n} \to_p L_{f,\chi_{[x,\infty)}}(\gamma, \sigma), \quad \frac{F_x(\hat{\mathbf{x}}(\mathbf{A}, \boldsymbol{\nu}, \mathbf{x}_0))}{n} \to_p M_{f,\chi_{[x,\infty)}}(\gamma, \sigma) \tag{7}$$

    *provided $L(x) = L_{f,\chi_{[x,\infty)}}(\gamma, \sigma)$ and $M(x) = M_{f,\chi_{[x,\infty)}}(\gamma, \sigma)$ are continuous at $x$.*

Notice that the above Theorem implies some parts of Theorem 1 in the paper: The claim of first bullet under assumption 1 is provided by part 1 of the above theorem. The claim of second bullet under assumption 1 is provided by part 3, noticing that $M$ and $L$ values in (4) correspond to the distribution of $\hat{X}$ and $W$ in the paper, respectively.

## 1.3 The Original LASSO

The error of the original LASSO cannot be characterized by Theorem 1. This case requires further restrictions on the choice of the random matrix. First, we remind the definition of the Restricted Isometry Property (RIP):

**Definition 2.** *Consider a $m \times n$ matrix $\mathbf{A}$.*

1. *For any natural number $k < n$, the RIP constant $\delta_k(\mathbf{A})$ is defined as the smallest numbers $\delta$, such that for any index subset $I \subset \{1, 2, \ldots, n\}$ with $|I| \leqslant k$*

$$1 - \delta \leqslant \sigma_{min}^2(\mathbf{A}_I) \leqslant \sigma_{max}^2(\mathbf{A}_I) \leqslant 1 + \delta \tag{8}$$

2. *We also define the admissible sparsity $M_{\mathrm{adm}}(\mathbf{A})$ as follows:*

$$M_{\mathrm{adm}}(\mathbf{A}) = \sup_k \frac{k[1 - \delta_k(\mathbf{A})]_+}{2n} \tag{9}$$

Then, we provide the following result:

**Theorem 2.** *Take g, $\boldsymbol{\nu}$ and $\mathbf{x}_0$ as in Theorem 1. Assume that $\mathbf{A}$ is a scaled-regular matrix with sub-Gaussian entries. Take $f(x) = \lambda|x|$, which yields to $f_n(\mathbf{x}) = \lambda\|\mathbf{x}\|_1$. Then,*

1. *The claim in Theorem 1.1 holds for $f(x) = \lambda|x|$. Moreover, $\|\hat{\mathbf{x}}(\mathbf{A}, \boldsymbol{\nu}, \mathbf{x}_0)\|_1/n \to_p M_{\lambda|x|,|x|}$.*

2. *Define*

$$M_0 = M_{f, \chi_{\mathbb{R}\setminus\{0\}}} = \Pr\left(\hat{x}_f\left(\frac{\hat{\beta}}{\hat{p}}, \hat{p}\Gamma + X\right) \neq 0\right) \tag{10}$$

   *as the "effective sparsity" of the LASSO estimate. If $M_0$ is strictly less than $M_{\mathrm{adm}}(\mathbf{A})$ with high probability[1], then the claims in Theorem 1.2 and 1.3 hold.*

The above result gives the remaining claims in Theorem 1 of the paper. Th calim of first bullet under assumption 2 is implied part 1 above, although the above holds under weaker assumptions. The claim of second bullet under assumption 2 is given in part 2 above, where we provide a lower bound, $\rho/2$ for $M_{\mathrm{adm}}$ by a standard argument given in [1].

## 1.4 Generalized Results

In Section (3.3 in the paper), we mentioned that the expressions in Theorem 1 and 2 are special cases of our more general discussion. Here, we mention these general results as an independent theorem:

**Theorem 3.**

1. *Suppose that in (2), $\mathbf{x}_0$ has i.i.d centered entries with distribution $\xi$ and $\mathbf{A}$ is $m \times n$. Moreover, $\{f_n\}$ and $\{g_n\}$ are regular and well-behaved sequences of functions, respectively, both with respect to $\xi$. Take $\boldsymbol{\nu}$ as an i.i.d Gaussian centered vector with distributions $\mathcal{N}(0, \sigma^2)$. Suppose that $m$ and $n$ grow such that $n = O(m)$. Take two scaled-regular random matrices $\mathbf{A}_1$ and $\mathbf{A}_2$. If the sequence $\Phi_n(\mathbf{A}_1, \boldsymbol{\nu}, \mathbf{x}_0)$ converges in probability to a value $C$, then $\Phi_n(\mathbf{A}_2, \boldsymbol{\nu}, \mathbf{x}_0)$ also converges in probability to $C$.*

2. *Consider a sequence $f^{(k)} : \mathbb{R} \to \mathbb{R}$ of strongly convex functions with bounded third derivatives uniformly converging to a function $f$. Then, the results of Theorem 1 holds for $f$.*

# 2 Definitions

**Definition 3. (conditions on $f$)**

1. *A sequence of three-times differentiable convex functions $\{f_n : \mathbb{R}^n \to \mathbb{R}\}$ for $n = 1, 2, \ldots$ is called smooth-regular with respect to a probability measure $\xi$ on $\mathbb{R}$, if there exist constants $C_1, C_2, \epsilon$ satisfying*

- *For every $n$,*

$$\int_{\mathbb{R}^n} \left( \frac{\|\nabla f_n(\mathbf{x})\|_2^2}{n} \right)^2 d^n\xi \leqslant C_1 \tag{11}$$

- *For every $n \in \mathbb{N}$, $\mathbf{x} \in \mathbb{R}^n$ and $\mathbf{s} = (s_1, s_2, \ldots, s_n)^T \in \mathbb{R}^n$,*

$$\sum_{\alpha,\beta} \frac{\partial^2 f_n}{\partial x_\alpha \partial x_\beta}(\mathbf{x}) s_\alpha s_\beta \geqslant \epsilon \sum_\alpha s_\alpha^2 \tag{12}$$

- *For every $n \in \mathbb{N}$, $\mathbf{x} \in \mathbb{R}^n$ and $\mathbf{s} = (s_1, s_2, \ldots, s_n)^T \in \mathbb{R}^n$,*

$$\sum_{\alpha,\beta,\gamma} \frac{\partial^3 f_n}{\partial x_\alpha \partial x_\beta \partial x_\gamma}(\mathbf{x}) s_\alpha s_\beta s_\gamma \leqslant C_2 \sum_\alpha |s_\alpha|^3 \tag{13}$$

2. *A sequence $\{f_n : \mathbb{R}^n \to \mathbb{R}\}$ of functions for $n = 1, 2, \ldots$ is regular if there exists a collection of functions $f_n^{(k)} : \mathbb{R}^n \to \mathbb{R}$ such that*

- *For every $k$, the sequence $\{f_n^{(k)}\}$ is smooth-regular.*
- *The sequence $\{\frac{1}{n} f_n^{(k)}\}$ converges uniformly in $\mathbf{x}$ and $n$ to $\{\frac{1}{n} f_n\}$ as $k \to \infty$. This means that for every $\epsilon > 0$ and sufficiently large $k$, the relation $|f_n^{(k)}(\mathbf{x}) - f_n(\mathbf{x})| < n\epsilon$ holds for every $n$ and $\mathbf{x} \in \mathbb{R}^n$.*

**Definition 4. (error risk function $g$)**

1. *A sequence of three-times differentiable convex function $g : \mathbb{R}^d \to \mathbb{R}$ is called well-behaved if there exist constants $C_1, \epsilon, C_2$, such that*

- *For every $n$,*

$$\frac{1}{n} \|\nabla g_n(0)\|_2^2 \leqslant C_1 \tag{14}$$

- *For every $n \in \mathbb{N}$, $\mathbf{x} \in \mathbb{R}^n$ and $\mathbf{s} = (s_1, s_2, \ldots, s_n)^T \in \mathbb{R}^n$,*

$$\sum_{\alpha,\beta} \frac{\partial^2 g_n}{\partial x_\alpha \partial x_\beta}(\mathbf{x}) s_\alpha s_\beta \leqslant \epsilon \sum_\alpha s_\alpha^2 \tag{15}$$

- *For every $n \in \mathbb{N}$, $\mathbf{x} \in \mathbb{R}^n$ and $\mathbf{s} = (s_1, s_2, \ldots, s_n)^T \in \mathbb{R}^n$,*

$$\sum_{\alpha,\beta,\gamma} \frac{\partial^3 g_n}{\partial x_\alpha \partial x_\beta \partial x_\gamma}(\mathbf{x}) s_\alpha s_\beta s_\gamma \leqslant C_2 \sum_\alpha |s_\alpha|^3 \tag{16}$$

2. *For any vector $\mathbf{x} \in \mathbb{R}^n$ and $x \in \mathbb{R}$, we define the empirical distribution $F_x(\mathbf{x})$ as*

$$F_x(\mathbf{x}) = \frac{1}{n} \sum_{k=1}^n \chi_{[x,\infty)}(x_k) \tag{17}$$

*where $\chi_S(x)$ is the characteristic function of $S$, which is one, when $x \in S$ and zero, otherwise.*

## 3   Proofs

### 3.1   A Technical Theorem

We start by proving another result, which will be useful in proving the previous theorems. We introduce

$$\Phi_{n,\rho}(\mathbf{A}, \boldsymbol{\nu}, \mathbf{x}_0) = \frac{1}{n} \min_{\mathbf{v}} \frac{1}{2} \|\boldsymbol{\nu} + \mathbf{A}\mathbf{v}\|_2^2 + f_n(\mathbf{v} + \mathbf{x}_0) + \rho g_n(\mathbf{v}) \tag{18}$$

Then, we show the following Lemma:

**Lemma 1.** *Suppose that in (1 in paper), $\mathbf{x}_0$ has i.i.d centered entries with distribution $\xi$ and $\mathbf{A}$ is $m \times n$. Assume that $f_n$ is a sooth-regular function with constants $C_1, \epsilon, C_2$ and $g_n$ is well-behaved with constants $(\tilde{C}_1, \tilde{\epsilon}, \tilde{C}_2)$. For $\rho > -\epsilon/\tilde{\epsilon}$ and any $\delta_2 > \delta_1 > 0$, if*

$$\lim_{n \to \infty} \Pr(|\Phi_{n,\rho}(\mathbf{A}_1, \boldsymbol{\nu}, \mathbf{x}_0) - C| > \delta_1) \to 0 \tag{19}$$

*then*

$$\lim_{n \to \infty} \Pr(|\Phi_{n,\rho}(\mathbf{A}_2, \boldsymbol{\nu}, \mathbf{x}_0) - C| > \delta_2) \to 0 \tag{20}$$

### 3.1.1 Proof of Lemma 1

The proof is based on the so-called Lindeberg's argument, which can be explained in the following steps:

**Step 1:** First, take any smooth function $h : \mathbb{R} \to \mathbb{R}$ with absolutely bounded derivatives of first and second order. It suffices to show that for such a function,

$$\lim_{n \to \infty} |\mathcal{E}\left(h\left(\Phi_{n,\rho}(\mathbf{A}_1, \boldsymbol{\nu}, \mathbf{x}_0)\right)\right) - \mathcal{E}\left(h\left(\Phi_{n,\rho}(\mathbf{A}_2, \boldsymbol{\nu}, \mathbf{x}_0)\right)\right)| \to 0 \tag{21}$$

since then, one may take a particular radially increasing smooth function $h_0$ with bounded first and second order derivatives, such that $h_0(x) = 0$ for $|x| \leqslant \delta_1$ and $1 < h_0(x) < 2$ for $|x| > \delta_2$. Note that by the assumption,

$$0 \leqslant \mathcal{E}(h_0(\Phi_{n,\rho}(\mathbf{A}_1, \boldsymbol{\nu}, \mathbf{x}_0) - C)) \leqslant 2\Pr(|\Phi_{n,\rho}(\mathbf{A}_1, \boldsymbol{\nu}, \mathbf{x}_0) - C| > \delta_1) \to 0 \tag{22}$$

Then, from (21) we have that

$$\mathcal{E}(h_0(\Phi_{n,\rho}(\mathbf{A}_2, \boldsymbol{\nu}, \mathbf{x}_0) - C)) \to 0 \tag{23}$$

On the other hand,

$$\Pr(|\Phi_{n,\rho}(\mathbf{A}_2, \boldsymbol{\nu}, \mathbf{x}_0) - C| > \delta_2) = \Pr(h(\Phi_{n,\rho}(\mathbf{A}_2, \boldsymbol{\nu}, \mathbf{x}_0) - C) > 1) \leqslant \mathcal{E}(h_0(\Phi_{n,\rho}(\mathbf{A}_2, \boldsymbol{\nu}, \mathbf{x}_0) - C)) \to 0 \tag{24}$$

where the last inequality is obtained by the Markov inequality.

**Step 2:** To show (21), take the intermediate matrices for $k = 1, \ldots, m - 1$

$$\mathbf{A}^{(k)} = \left[\mathbf{a}_{1,1} \, \mathbf{a}_{1,2} \ldots \mathbf{a}_{1,k} \, \mathbf{a}_{2,k+1} \, \mathbf{a}_{2,k+2} \ldots \mathbf{a}_{2,m}\right]^T \tag{25}$$

where $\mathbf{a}_{i,k} \in \mathbb{R}^n$ are the transpose of the $k^{\text{th}}$ row of $\mathbf{A}_i$. Define $\mathbf{A}_2 = \mathbf{A}^{(0)}$ and $\mathbf{A}_1 = \mathbf{A}^{(m)}$ and

$$\mathbf{R}^{(k)} = \left[\mathbf{a}_{1,1} \, \mathbf{a}_{1,2} \ldots \mathbf{a}_{1,k} \, \mathbf{a}_{2,k+2} \, \mathbf{a}_{2,k+3} \ldots \mathbf{a}_{2,m}\right]^T \tag{26}$$

for $k = 1, 2, \ldots, m - 1$, with

$$\mathbf{R}^{(0)} = \left[\mathbf{a}_{2,2} \, \mathbf{a}_{2,3} \, \ldots \, \mathbf{a}_{2,m}\right]^T \tag{27}$$

It is now easy to see that defining

$$\Theta_{n,\rho}(\mathbf{R}, \mathbf{a}, \boldsymbol{\nu}, \nu, \mathbf{x}) = \frac{1}{n} \min_{\mathbf{v}} \frac{1}{2} \|\boldsymbol{\nu} + \mathbf{R}\mathbf{v}\|_2^2 + \frac{1}{2}(\nu + \mathbf{a}^T \mathbf{v})^2 + f_n(\mathbf{v} + \mathbf{x}) + \rho g_n(\mathbf{v}) \tag{28}$$

and

$$\boldsymbol{\nu}_{-k} = \left[\nu_1 \, \nu_2 \, \ldots \nu_{k-1} \, \nu_{k+1} \, \ldots \nu_m\right]^T \tag{29}$$

we have that

$$\Phi_{n,\rho}(\mathbf{A}^{(k)}, \boldsymbol{\nu}, \mathbf{x}) = \Theta_{n,\rho}(\mathbf{R}^{(k)}, \mathbf{a}_{2,k}, \boldsymbol{\nu}_{-(k+1)}, \nu_{k+1}, \mathbf{x}) \tag{30}$$

and

$$\Phi_{n,\rho}(\mathbf{A}^{(k+1)}, \boldsymbol{\nu}, \mathbf{x}) = \Theta_{n,\rho}(\mathbf{R}^{(k)}, \mathbf{a}_{1,k+1}, \boldsymbol{\nu}_{-(k+1)}, \nu_{k+1}, \mathbf{x}) \tag{31}$$

Now, we use the fact that

$$\left|\mathcal{E}\left(h\left(\Phi_{n,\rho}(\mathbf{A}_1, \boldsymbol{\nu}, \mathbf{x})\right)\right) - \mathcal{E}\left(h\left(\Phi_{n,\rho}(\mathbf{A}_2, \boldsymbol{\nu}, \mathbf{x})\right)\right)\right| =$$

$$\left|\sum_{k=0}^{m-1} \mathcal{E}\left(h\left(\Phi_{n,\rho}(\mathbf{A}^{(k+1)}, \boldsymbol{\nu}, \mathbf{x})\right)\right) - \mathcal{E}\left(h\left(\Phi_{n,\rho}(\mathbf{A}^{(k)}, \boldsymbol{\nu}, \mathbf{x})\right)\right)\right| \leqslant$$

$$\sum_{k=0}^{m-1} \left|\mathcal{E}\left(h\left(\Phi_{n,\rho}(\mathbf{A}^{(k+1)}, \boldsymbol{\nu}, \mathbf{x})\right)\right) - \mathcal{E}\left(h\left(\Phi_{n,\rho}(\mathbf{A}^{(k)}, \boldsymbol{\nu}, \mathbf{x})\right)\right)\right| \tag{32}$$

Furthermore,

$$\left| \mathcal{E}\left(h\left(\Phi_{n,\rho}(\mathbf{A}^{(k+1)}, \boldsymbol{\nu}, \mathbf{x})\right)\right) - \mathcal{E}\left(h\left(\Phi_{n,\rho}(\mathbf{A}^{(k)}, \boldsymbol{\nu}, \mathbf{x})\right)\right)\right| =$$

$$\left| \mathcal{E}\left(h\left(\Theta_{n,\rho}(\mathbf{R}^{(k)}, \mathbf{a}_{2,k}, \boldsymbol{\nu}_{-(k+1)}, \nu_{k+1}, \mathbf{x})\right)\right) - \mathcal{E}\left(h\left(\Theta_{n,\rho}(\mathbf{R}^{(k)}, \mathbf{a}_{1,k+1}, \boldsymbol{\nu}_{-(k+1)}, \nu_{k+1}, \mathbf{x})\right)\right)\right| =$$

$$\left| \mathcal{E}\left(h\left(\Theta_{n,\rho}(\mathbf{R}^{(k)}, \mathbf{a}_{2,k}, \boldsymbol{\nu}_{-(k+1)}, \nu_{k+1}, \mathbf{x})\right)\right) - \mathcal{E}\left(h\left(\Phi_{n,\rho}(\mathbf{R}^{(k)}, \boldsymbol{\nu}_{-(k+1)}, \mathbf{x})\right)\right) - \right.$$

$$\left. \mathcal{E}\left(h\left(\Theta_{n,\rho}(\mathbf{R}^{(k)}, \mathbf{a}_{1,k+1}, \boldsymbol{\nu}_{-(k+1)}, \nu_{k+1}, \mathbf{x})\right)\right) + \mathcal{E}\left(h\left(\Phi_{n,\rho}(\mathbf{R}^{(k)}, \boldsymbol{\nu}_{-(k+1)}, \mathbf{x})\right)\right)\right| \qquad (33)$$

For the sake of simplicity, let us define

$$\Delta_{2,k} = \mathcal{E}\left(h\left(\Theta_{n,\rho}(\mathbf{R}^{(k)}, \mathbf{a}_{2,k}, \boldsymbol{\nu}_{-(k+1)}, \nu_{k+1}, \mathbf{x})\right) - h\left(\Phi_{n,\rho}(\mathbf{R}^{(k)}, \boldsymbol{\nu}_{-(k+1)}, \mathbf{x})\right)\right) \qquad (34)$$

and

$$\Delta_{2,k} = \mathcal{E}\left(h\left(\Theta_{n,\rho}(\mathbf{R}^{(k)}, \mathbf{a}_{1,k+1}, \boldsymbol{\nu}_{-(k+1)}, \nu_{k+1}, \mathbf{x})\right) - h\left(\Phi_{n,\rho}(\mathbf{R}^{(k)}, \boldsymbol{\nu}_{-(k+1)}, \mathbf{x})\right)\right) \qquad (35)$$

Then, (33) can be written as

$$\left| \mathcal{E}\left(h\left(\Phi_{n,\rho}(\mathbf{A}^{(k+1)}, \boldsymbol{\nu}, \mathbf{x})\right)\right) - \mathcal{E}\left(h\left(\Phi_{n,\rho}(\mathbf{A}^{(k)}, \boldsymbol{\nu}, \mathbf{x})\right)\right)\right| = |\Delta_{2,k} - \Delta_{1,k}| \qquad (36)$$

Now, note that since $|h''(x)| \leqslant H_2$ for a proper value of $H_2$, we have that

$$|h(x) - h(y) - h'(x)(y - x)| \leqslant H_2(y - x)^2 \qquad (37)$$

which leads to

$$\left| \Delta_{i,k} - \mathcal{E}\left[h'\left(\Phi_{n,\rho}(\mathbf{R}^{(k)}, \boldsymbol{\nu}_{-(k+1)}, \mathbf{x})\right)\Delta'_{i,k}\right]\right| \leqslant H_2 \mathcal{E}\left[\left(\Delta'_{i,k}\right)^2\right] \qquad (38)$$

where for $i = 1, 2$

$$\Delta'_{i,k} = \Theta_{n,\rho}(\mathbf{R}^{(k)}, \mathbf{a}_{i,k+2-i}, \boldsymbol{\nu}_{-(k+1)}, \nu_{k+1}, \mathbf{x}) - \Phi_{n,\rho}(\mathbf{R}^{(k)}, \boldsymbol{\nu}_{-(k+1)}, \mathbf{x}) \qquad (39)$$

Thus,

$$|\Delta_{2,k} - \Delta_{1,k}| \leqslant \left| \mathcal{E}\left[h'\left(\Phi_{n,\rho}(\mathbf{R}^{(k)}, \boldsymbol{\nu}_{-(k+1)}, \mathbf{x})\right)\left(\Delta'_{2,k} - \Delta'_{1,k}\right)\right]\right| + H_2 \sum_{i=1,2} \mathcal{E}\left[\left(\Delta'_{i,k}\right)^2\right] \quad (40)$$

Finally, note that

$$\mathcal{E}\left[h'\left(\Phi_{n,\rho}(\mathbf{R}^{(k)}, \boldsymbol{\nu}_{-(k+1)}, \mathbf{x})\right)\Delta'_{i,k}\right] =$$

$$\mathcal{E}_{\mathbf{R}^{(k)}, \boldsymbol{\nu}_{-(k+1)}, \mathbf{x}}\left[h'\left(\Phi_{n,\rho}(\mathbf{R}^{(k)}, \boldsymbol{\nu}_{-(k+1)}, \mathbf{x})\right)\mathcal{E}_{\mathbf{a}_{i,k+2-i}, \nu_{k+1}}\left(\Delta'_{i,k} \mid \mathbf{R}^{(k)}, \boldsymbol{\nu}_{-(k+1)}, \mathbf{x}\right)\right],$$

$$\mathcal{E}\left[\left(\Delta'_{i,k}\right)^2\right] = \mathcal{E}_{\mathbf{R}^{(k)}, \boldsymbol{\nu}_{-(k+1)}, \mathbf{x}}\left[\mathcal{E}_{\mathbf{a}_{i,k+2-i}, \nu_{k+1}}\left[\left(\Delta'_{i,k}\right)^2 \mid \mathbf{R}^{(k)}, \boldsymbol{\nu}_{-(k+1)}, \mathbf{x}\right]\right] \qquad (41)$$

We show in the sequel that there exists a constant $Q$ such that

$$|\Delta_{2,k} - \Delta_{1,k}| \leqslant Q m^{-\frac{5}{4}} \qquad (42)$$

Then, due to (32), we get that

$$\left| \mathcal{E}\left(h\left(\Phi_{n,\rho}(\mathbf{A}_1, \boldsymbol{\nu}, \mathbf{x})\right)\right) - \mathcal{E}\left(h\left(\Phi_{n,\rho}(\mathbf{A}_2, \boldsymbol{\nu}, \mathbf{x})\right)\right)\right| \leqslant \sum_k |\Delta_{2,k} - \Delta_{1,k}| \leqslant Q m^{-\frac{1}{4}} \to 0 \qquad (43)$$

which proves the result.

**Step 3:** To obtain (42), we analyze each term in (40) separately. This means that we nead to calculate the leading terms of the statistics of $\Delta'_{i,k}$. However, as (41) suggests, $\mathbf{R}^{(k)}, \boldsymbol{\nu}_{-(k+1)}, \mathbf{x}$ is assumed to be deterministic and limits only for the inner expectations in (41) are calculated, which automatically leads to bounds for the outer expectations. Then, we denote $\mathbf{R}^{(k)} = \mathbf{R}, \boldsymbol{\nu}_{-(k+1)} = \boldsymbol{\mu}$ as the analysis is for a fixed $k$ and assume that $\mathbf{R}, \boldsymbol{\mu}$ are deterministic values. As the analysis for $i = 1, 2$ are symmetric, we drop the index $i$ and do the analysis for an i.i.d random vector $\mathbf{a}$ with a regular

distribution, as well as a Gaussian random variable $\nu$ which will be later replaced by $\mathbf{a}_{i,k-i+2}$ for $i = 1, 2$ and $\nu_k$, respectively. This means that we analyze the statistics of

$$\Delta' = \Theta_{n,\rho}(\mathbf{R}, \mathbf{a}, \boldsymbol{\mu}, \nu, \mathbf{x}) - \Phi_{n,\rho}(\mathbf{R}, \boldsymbol{\mu}, \mathbf{x}) \tag{44}$$

Note that $\Phi_{n,\rho}(\mathbf{R}, \boldsymbol{\mu}, \mathbf{x}) = \Theta_{n,\rho}(\mathbf{R}, \mathbf{a} = 0, \boldsymbol{\mu}, \nu = 0, \mathbf{x})$. Thus, $\Delta'$ is a perturbation in $\Theta$ and can be calculated by standard perturbation theory. Denote by $\mathbf{w}_\rho(\mathbf{A}, \nu, \mathbf{x})$ the optimal point of (18) and take $\hat{\mathbf{v}} = \mathbf{w}_\rho(\mathbf{R}, \boldsymbol{\mu}, \mathbf{x})$. Note that we also drop the indexes $n, \rho$ for simplicity. Define

$$\Psi(\mathbf{R}, \mathbf{a}, \boldsymbol{\mu}, \nu) = \frac{1}{m} \min_{\mathbf{v}} \frac{1}{2}\|\boldsymbol{\mu} + \mathbf{R}\mathbf{v}\|_2^2 + \frac{1}{2}(\nu + \mathbf{a}^T\mathbf{v})^2 +$$

$$f(\hat{\mathbf{v}} + \mathbf{x}) + \rho g(\hat{\mathbf{v}}) + \boldsymbol{\gamma}^T(\mathbf{v} - \hat{\mathbf{v}}) + \frac{1}{2}(\mathbf{v} - \hat{\mathbf{v}})^T\mathbf{H}(\mathbf{v} - \hat{\mathbf{v}}) \tag{45}$$

where

$$\boldsymbol{\gamma} = \nabla f(\hat{\mathbf{v}} + \mathbf{x}) + \rho\nabla g(\hat{\mathbf{v}}), \quad \mathbf{H} = \frac{\partial^2 f(\hat{\mathbf{v}} + \mathbf{x})}{\partial\mathbf{x}\partial\mathbf{x}^T} + \rho\frac{\partial^2 g(\hat{\mathbf{v}})}{\partial\mathbf{x}\partial\mathbf{x}^T} \tag{46}$$

Defining $\Delta\mathbf{v} = \mathbf{v} - \hat{\mathbf{v}}$, the expression in (45) can be equivalently written as

$$\Psi(\mathbf{R}, \mathbf{a}, \boldsymbol{\mu}, g) = \frac{1}{m} \min_{\Delta\mathbf{v}} \frac{1}{2}\|\mathbf{z} + \mathbf{R}\Delta\mathbf{v}\|_2^2 + \frac{1}{2}(\gamma + \mathbf{a}^T\Delta\mathbf{v})^2 +$$

$$f(\hat{\mathbf{v}} + \mathbf{x}) + \rho g(\hat{\mathbf{v}}) + \boldsymbol{\gamma}^T\Delta\mathbf{v} + \frac{1}{2}\Delta\mathbf{v}^T\mathbf{H}\Delta\mathbf{v} \tag{47}$$

where

$$\gamma = \nu + \mathbf{a}^T\hat{\mathbf{v}}, \quad \mathbf{z} = \boldsymbol{\mu} + \mathbf{R}\hat{\mathbf{v}} \tag{48}$$

Also, note that since $\hat{\mathbf{v}}$ is the optimal solution of (18), it satisfies

$$\mathbf{R}^T\mathbf{z} + \boldsymbol{\gamma} = \mathbf{0} \tag{49}$$

Thus, defining $\Delta\tilde{\mathbf{v}}$ as the optimal point of (47), we have that

$$\Delta\tilde{\mathbf{v}} = -\gamma(\boldsymbol{\Omega} + \mathbf{a}\mathbf{a}^T)^{-1}\mathbf{a} = \frac{-\gamma\boldsymbol{\Omega}^{-1}\mathbf{a}}{1 + \mathbf{a}^T\boldsymbol{\Omega}^{-1}\mathbf{a}} \tag{50}$$

where

$$\boldsymbol{\Omega} = \mathbf{R}^T\mathbf{R} + \mathbf{H} \tag{51}$$

and

$$\Theta(\mathbf{R}, \mathbf{a}, \boldsymbol{\mu}, \nu, \mathbf{x}) - \Phi(\mathbf{R}, \boldsymbol{\mu}, \mathbf{x}) = \frac{1}{m}\left(\frac{1}{2}\gamma^2 - \frac{1}{2}\gamma^2\mathbf{a}^T(\boldsymbol{\Omega} + \mathbf{a}\mathbf{a}^T)^{-1}\mathbf{a}\right) = \frac{\frac{1}{2m}\gamma^2}{1 + \mathbf{a}^T\boldsymbol{\Omega}^{-1}\mathbf{a}} \tag{52}$$

we also define

$$\tilde{\Delta} = \tilde{\Delta}(\mathbf{R}, \mathbf{a}, \boldsymbol{\nu}, \nu, \mathbf{x}) = \Psi(\mathbf{R}, \mathbf{a}, \boldsymbol{\nu}, \nu, \mathbf{x}) - \Phi(\mathbf{R}, \boldsymbol{\nu}, \mathbf{x}) = \frac{\frac{1}{2m}\gamma^2}{1 + \mathbf{a}^T\boldsymbol{\Omega}^{-1}\mathbf{a}} \tag{53}$$

In Lemma 7, we will show that there exists a value $\eta_1 > 0$, such that:

$$|\Delta' - \tilde{\Delta}| \leqslant \frac{1}{2m}\gamma^2\chi_{\{\|\tilde{\mathbf{v}}\|_3^3 > \eta_1\}} + \frac{\epsilon\epsilon_1}{m}\|\tilde{\mathbf{v}}\|_3^3\chi_{\{\|\tilde{\mathbf{v}}\|_3^3 < \eta_1\}} \tag{54}$$

Then from Lemma 6, we conclude that

$$|\mathcal{E}_{\mathbf{a},\nu}(\Delta') - \mathcal{E}_{\mathbf{a},\nu}(\tilde{\Delta})| \leqslant \mathcal{E}_{\mathbf{a},\nu}|\Delta' - \tilde{\Delta}| \leqslant \mathcal{E}_{\mathbf{a},\nu}(\frac{1}{2m}\gamma^2\chi_{\{\|\tilde{\mathbf{v}}\|_3^3 > \eta_1\}}) + \frac{\epsilon\epsilon_1}{m}\mathcal{E}_{\mathbf{a},\nu}(\|\tilde{\mathbf{v}}\|_3^3) \leqslant$$

$$\frac{1}{2m}\sqrt{\Pr(\|\tilde{\mathbf{v}}\|_3^3 > \eta_1)\mathcal{E}_{\mathbf{a},\nu}(\gamma^4)} + \frac{\epsilon\epsilon_1}{m}\mathcal{E}_{\mathbf{a},\nu}(\|\tilde{\mathbf{v}}\|_3^3) \leqslant$$

$$\frac{1}{2m}(L_1 + K_1\frac{\|\hat{\mathbf{v}}\|_2^2}{m})\sqrt{\frac{\sqrt{L_2 + K_2\frac{\|\hat{\mathbf{v}}\|_2^6}{m^3}}}{\eta\epsilon^3\sqrt{m}}} + \frac{\epsilon\epsilon_1}{m}\frac{\sqrt{L_2 + K_2\frac{\|\hat{\mathbf{v}}\|_2^6}{m^3}}}{\epsilon^3\sqrt{m}} \tag{55}$$

Combining this with (89) in Lemma 6, results in

$$\left|\mathcal{E}_{\mathbf{a},\nu}(\Delta') - \frac{\frac{1}{2m}(1 + \frac{\|\hat{\mathbf{v}}\|_2^2}{m})}{1 + \frac{\text{Tr}(\boldsymbol{\Omega}^{-1})}{m}}\right| \leqslant m^{-\frac{3}{2}}\left(L_1' + K_1'\left(\frac{\|\hat{\mathbf{v}}\|_2^2}{m}\right)^{\frac{3}{2}}\right) + m^{-\frac{5}{4}}\left(L_2' + K_2'\left(\frac{\|\hat{\mathbf{v}}\|_2^2}{m}\right)^{\frac{7}{4}}\right) \leqslant$$

$$m^{-\frac{5}{4}}\left(L' + K'\left(\frac{\|\hat{\mathbf{v}}\|_2^2}{m}\right)^2\right) \tag{56}$$

For proper choice of $L'_1, L'_2, L'$ and $K'_1, K'_2, K'$. We also conclude from (116) that there exists constants $K''$ and $L''$ such that

$$\mathcal{E}_{\mathbf{a},\nu}\left((\Delta')^2\right) \leqslant \frac{1}{4m^2}\mathcal{E}_{\mathbf{a},\nu}(\gamma^4) = \frac{(1+\frac{\|\hat{\mathbf{v}}\|_2^2}{m})^2}{4m^2} \leqslant m^{-2}\left(L'' + K''\left(\frac{\|\hat{\mathbf{v}}\|_2^2}{m}\right)^2\right) \quad (57)$$

Now, using the definition of $\Delta'_{i,k}$, we conclude that

$$\left|\mathcal{E}_{\mathbf{a}_{2,k},\nu_{-(k+1)}}(\Delta'_{2,k}) - \mathcal{E}_{\mathbf{a}_{1,k+1},\nu_{-(k+1)}}(\Delta'_{1,k})\right| \leqslant 2m^{-\frac{5}{4}}\left(L' + K'\left(\frac{\|\hat{\mathbf{v}}_k\|_2^2}{m}\right)^2\right) \quad (58)$$

where $\hat{\mathbf{v}}_k = \mathbf{w}_\rho(\mathbf{R}_k, \nu_{-(k+1)}, \mathbf{x})$. Using (40), (41) and noting that $h'(x) \leqslant D_1$ for a proper value of $D_1$, we have that

$$|\Delta_{2,k} - \Delta_{1,k}| \leqslant$$

$$2D_1 m^{-\frac{5}{4}}\left(L' + K'\mathcal{E}\left(\left(\frac{\|\hat{\mathbf{v}}_k\|_2^2}{m}\right)^2\right)\right) + 2D_2 m^{-2}\left(L'' + K''\mathcal{E}\left(\left(\frac{\|\hat{\mathbf{v}}\|_2^2}{m}\right)^2\right)\right) \leqslant$$

$$m^{-\frac{5}{4}}\left(\bar{L} + \bar{K}\mathcal{E}\left(\left(\frac{\|\hat{\mathbf{v}}_k\|_2^2}{m}\right)^2\right)\right) \quad (59)$$

for a proper choice of constants $\bar{L}, \bar{K}$. Note that by Lemma 5 we have that

$$\mathcal{E}\left(\left(\frac{\|\hat{\mathbf{v}}_k\|_2^2}{m}\right)^2\right) \leqslant \mathcal{E}\left(\left(\frac{6(\|\mathbf{R}_k^T\nu_{-(k+1)}\|_2^2 + \|\nabla f(\mathbf{x})\|_2^2 + \rho^2\|\nabla g(\mathbf{0})\|_2^2)}{\epsilon^2 m}\right)^2\right) \leqslant$$

$$\frac{108}{\epsilon^4}\mathcal{E}\left(\left(\frac{\|\mathbf{R}_k^T\nu_{-(k+1)}\|_2^2)}{m}\right)^2\right) + \frac{108}{\epsilon^4}\mathcal{E}\left(\left(\frac{\|\nabla f(\mathbf{x})\|_2^2}{m}\right)^2\right) + \frac{108\rho^4}{\epsilon^4}\left(\frac{\|\nabla g(\mathbf{0})\|_2^2}{m}\right)^2 \quad (60)$$

Thus, due to Lemma 4, there exists a constant $Q$ such that

$$|\Delta_{2,k} - \Delta_{1,k}| \leqslant Qm^{-\frac{5}{4}} \quad (61)$$

Then, due to (32), we get that

$$\left|\mathcal{E}\left(h\left(\Phi(\mathbf{A}_1, \mathbf{g})\right)\right) - \mathcal{E}\left(h\left(\Phi(\mathbf{A}_2, \mathbf{g})\right)\right)\right| \leqslant \sum_k |\Delta_{2,k} - \Delta_{1,k}| \leqslant Qm^{-\frac{1}{4}} \to 0 \quad (62)$$

which completes the proof.

**Lemma 2.** *For every $\alpha > 0$, there exists numbers $\eta_1, \epsilon_1 > 0$, such that if $0 < x < \eta_1$, then there exists $0 < r \leqslant \epsilon_1 x$ satisfying*

$$4r^{\frac{3}{2}} + 5x < \alpha r \quad (63)$$

*Proof.* Take $\delta > 1$ and

$$\eta_0 = \max_{r>0} F(r) \quad (64)$$

where $F(r) = (\alpha r - 4r^{\frac{3}{2}})/5$ and denote by $r_0$ its optimal point. Define $\eta_1 = \eta_0/\delta$. Note that $\eta_0, r_0 > 0$, since $F'(0) = \alpha/5 > 0$. Moreover, $F(0) = 0$. Thus, by the mean value theorem, for any $0 < x < \eta_1$ there exists a point $0 < r < r_0$, such that $F(r) = \delta x > x$, thus satisfying (63). Define

$$\epsilon_0 = \max_{0<r<r_0} \frac{r}{F(r)} = \frac{5}{\min_{0<r<r_0} \alpha - 4\sqrt{r}} = \frac{5}{\alpha - 4\sqrt{r_0}} = \frac{r_0}{F(r_0)} > 0 \quad (65)$$

and $\epsilon_1 = \delta\epsilon_0$. Now,

$$\frac{r}{\delta x} = \frac{r}{F(r)} \leqslant \epsilon_0 \to r \leqslant \epsilon_0\delta x = \epsilon_1 x \quad (66)$$

$\square$

**Lemma 3.** *Suppose that the $m-$dimensional vector $\sqrt{m}\mathbf{h}$ has i.i.d regular entries. For any $m \times m$ matrix $\mathbf{S}$, vector $\mathbf{z}$ and $d = 1, 2, 3$*

$$\mathcal{E}(|\mathbf{z}^T\mathbf{h}|^{2d}) \leqslant \frac{(2d)!}{d! \times 2^d}\frac{\|\mathbf{z}\|^{2d}}{m}\mathcal{E}(h_1^{2d}) \tag{67}$$

$$\mathcal{E}(\mathbf{h}^T\mathbf{Sh}) = \frac{Tr(\mathbf{S})}{m} \tag{68}$$

$$\mathcal{E}(|\mathbf{h}^T\mathbf{Sh}|^2) \leqslant \left(\frac{Tr(\mathbf{S}^2)}{m^2}\right)^2 + 2\frac{Tr(\mathbf{S}^4)}{m^4}\mathcal{E}(z^4) \tag{69}$$

*Proof.* For the first part, note that

$$\mathcal{E}(|\mathbf{z}^T\mathbf{h}|^{2d}) = \sum \mathcal{E}(z_{\alpha_1}z_{\alpha_2}\ldots z_{\alpha_d}h_{\alpha_1}h_{\alpha_2}\ldots h_{\alpha_d}) \tag{70}$$

Note that only the well-paired terms contribute to the summation. Then, using the union bound and noting that $\mathcal{E}(z^{2l}) \leqslant \mathcal{E}(z^{2d})^{l/d}$ for any $l \leqslant d$, we obtain the results. For the next part, not that

$$\mathcal{E}(\mathbf{h}^T\mathbf{Sh}) = Tr(\mathbf{S}\mathcal{E}(\mathbf{hh}^T)) = \frac{Tr(\mathbf{S})}{m} \tag{71}$$

and finally,

$$\mathcal{E}(|\mathbf{h}^T\mathbf{Sh}|^2) = \sum \mathcal{E}(h_{\alpha_1}h_{\alpha_2}h_{\beta_1}h_{\beta_2})S_{\alpha_1,\beta_1}S_{\alpha_2,\beta_2} \tag{72}$$

The well-paired terms only contribute to the summation. Then, using the union bound and noting that $\mathcal{E}(z^4) \geqslant \mathcal{E}(z^2)^2$, we obtain the result □

**Lemma 4.** *By the definitions and conditions above, we have that*

$$\frac{1}{m^2}\mathcal{E}\left(\|\mathbf{R}_k^T\boldsymbol{\nu}_{-k+1}\|_2^4\right) \leqslant D \tag{73}$$

*for a proper choice of $D$, independent of $m$ and $n$.*

*Proof.* Note that

$$(\mathbf{R}_k^T\boldsymbol{\nu}_{-k+1})^T = \sum_{l<k}\mathbf{a}_{1,l}\nu_l + \sum_{l>k}\mathbf{a}_{2,l}\nu_l = \sum_{l\neq k}\mathbf{b}_k \tag{74}$$

where $\mathbf{b}_k = \mathbf{a}_{l,1}\nu_l$ if $l < k$ or $\mathbf{b}_k = \mathbf{a}_{l,2}\nu_l$ otherwise. Thus,

$$\mathcal{E}\left(\|\mathbf{R}_k^T\boldsymbol{\nu}_{-k+1}\|_2^4\right) = \mathcal{E}\left(\sum_{\alpha_1,\alpha_2,\beta_1,\beta_2}\mathbf{b}_{\alpha_1}^T\mathbf{b}_{\beta_1}\mathbf{b}_{\alpha_2}^T\mathbf{b}_{\beta_2}\right) \tag{75}$$

Note that $\mathbf{b}_a$ and $\mathbf{b}_b$ are centered and independent if $a \neq b$. Thus,

$$\mathcal{E}\left(\|\mathbf{R}_k^T\boldsymbol{\nu}_{-k+1}\|_2^4\right) \leqslant$$

$$\mathcal{E}\left(\sum_{\alpha_1,\alpha_2}\mathbf{b}_{\alpha_1}^T\mathbf{b}_{\alpha_1}\mathbf{b}_{\alpha_2}^T\mathbf{b}_{\alpha_2}\right) + 2\mathcal{E}\left(\sum_{\alpha_1,\beta_1}\mathbf{b}_{\alpha_1}^T\mathbf{b}_{\beta_1}\mathbf{b}_{\alpha_1}^T\mathbf{b}_{\beta_1}\right) + \mathcal{E}\left(\sum_{\alpha_1}\mathbf{b}_{\alpha_1}^T\mathbf{b}_{\alpha_1}\mathbf{b}_{\alpha_1}^T\mathbf{b}_{\alpha_1}\right) =$$

$$\left(\sum_{\alpha}\mathcal{E}\left(\|\mathbf{b}_\alpha\|_2^2\right)\right)^2 + 2\sum_{\alpha,\beta}Tr\left(\mathcal{E}\left(\mathbf{b}_\alpha\mathbf{b}_\alpha^T\right)\mathcal{E}\left(\mathbf{b}_\beta^T\mathbf{b}_\beta\right)\right) + \sum_{\alpha_1}\mathcal{E}\left(\|\mathbf{b}_\alpha\|_2^4\right) \tag{76}$$

On the other hand, $\mathcal{E}(\mathbf{b}_\alpha\mathbf{b}_\alpha^T) = \mathbf{I}/m$ and $\mathcal{E}(\|\mathbf{b}_\alpha\|_2^2) = 1$. Furthermore,

$$\mathcal{E}(\|\mathbf{b}_\alpha\|_2^4) = \mathcal{E}(\nu_\alpha^4)\mathcal{E}(\|\mathbf{a}\|_2^4) \leqslant D_1\mathcal{E}\left(\left(\sum_\mu a_\mu^2\right)^2\right) \leqslant D_1\left(\sum_\mu \mathcal{E}a_\mu^4 + 2\left(\sum_\mu \mathcal{E}a_\mu^2\right)^2\right) \leqslant D_1(D_2+2) \tag{77}$$

Thus,

$$\frac{1}{m^2}\mathcal{E}\left(\|\mathbf{R}_k^T\boldsymbol{\nu}_{-k+1}\|_2^4\right) \leqslant \left(\frac{n}{m}\right)^2 + \frac{2n^2}{m^3} + \frac{D_1(D_2+2)}{m} \leqslant D \tag{78}$$

For a proper choice of $D$. □

**Lemma 5.** *The norm of* $\mathbf{w}_\rho(\mathbf{R}, \boldsymbol{\nu}, \mathbf{x})$ *is bounded by*

$$\|\mathbf{w}_\rho(\mathbf{R}, \boldsymbol{\nu}, \mathbf{x})\|_2^2 \leqslant \frac{6(\|\mathbf{R}^T\boldsymbol{\nu}\|_2^2 + \|\nabla f(\mathbf{x})\|_2^2 + \rho^2\|\nabla g(\mathbf{0})\|)}{\epsilon^2} \tag{79}$$

*Proof.* To see this, take an arbitrary point $\mathbf{v}$ and define

$$\eta(t) = \frac{1}{n}\left(\frac{1}{2}\|\boldsymbol{\nu} + \mathbf{R}\mathbf{v}t\|_2^2 + f(\mathbf{v}t + \mathbf{x}) + \rho g(\mathbf{v}t)\right) \tag{80}$$

Note that

$$\eta(0) = \frac{1}{n}\left(\frac{1}{2}\|\boldsymbol{\nu}\|_2^2 + f(\mathbf{x}) + \rho g(\mathbf{0})\right) \tag{81}$$

and

$$\eta'(0) = \frac{1}{n}\left((\mathbf{R}^T\boldsymbol{\nu} + \nabla f(\mathbf{x}) + \rho\nabla g(\mathbf{0}))^T\mathbf{v}\right) \tag{82}$$

Moreover,

$$\eta''(t) = \frac{1}{m}\left(\mathbf{v}^T\mathbf{R}^T\mathbf{R}\mathbf{v} + \sum_{\alpha,\beta}\frac{\partial^2 f}{\partial x_\alpha \partial x_\beta}v_\alpha v_\beta + \rho\sum_{\alpha,\beta}\frac{\partial^2 g}{\partial x_\alpha \partial x_\beta}v_\alpha v_\beta\right) \geqslant \epsilon\frac{\|\mathbf{v}\|_2^2}{m} \tag{83}$$

where we use the second property of $f$ and convexity of $g$. Integrating the above, we obtain that

$$\eta'(t) \geqslant \frac{1}{m}\left((\mathbf{R}^T\boldsymbol{\nu} + \nabla f(\mathbf{x}) + \rho\nabla g(\mathbf{0}))^T\mathbf{v} + \epsilon\|\mathbf{v}\|_2^2 t\right) \tag{84}$$

and by integration again, we obtain that

$$\eta(t) \geqslant \frac{1}{m}\left(\frac{1}{2}\|\boldsymbol{\nu}\|_2^2 + f(\mathbf{x}) + \rho g(\mathbf{0}) + (\mathbf{R}^T\boldsymbol{\nu} + \nabla f(\mathbf{x}) + \nabla g(\mathbf{0}))^T\mathbf{v}t + \epsilon\|\mathbf{v}\|_2^2\frac{t^2}{2}\right) \tag{85}$$

Setting $t = 1$, we obtain that

$$\frac{1}{m}\left(\frac{1}{2}\|\boldsymbol{\nu} + \mathbf{R}\mathbf{v}\|_2^2 + f(\mathbf{v} + \mathbf{x}) + \rho g(\mathbf{v})\right) \geqslant$$

$$\frac{1}{m}\left(\frac{1}{2}\|\boldsymbol{\nu}\|_2^2 + f(\mathbf{x}) + \rho g(\mathbf{0}) + (\mathbf{R}^T\boldsymbol{\nu} + \nabla f(\mathbf{x}) + \rho\nabla g(\mathbf{0}))^T\mathbf{v} + \frac{\epsilon\|\mathbf{v}\|_2^2}{2}\right) \tag{86}$$

It is now clear that

$$\mathbf{w}_\rho(\mathbf{R}, \boldsymbol{\nu}, \mathbf{x}) \in \left\{\mathbf{v} \mid \frac{1}{2}\|\boldsymbol{\nu} + \mathbf{R}\mathbf{v}\|_2^2 + f(\mathbf{v} + \mathbf{x}) + \rho g(\mathbf{v}) \leqslant \frac{1}{2}\|\boldsymbol{\nu}\|_2^2 + f(\mathbf{x}) + \rho g(\mathbf{0})\right\}$$

$$\subseteq \left\{\mathbf{v} \mid (\mathbf{R}^T\boldsymbol{\nu} + \nabla f(\mathbf{x}) + \rho\nabla g(\mathbf{0}))^T\mathbf{v} + \frac{\epsilon\|\mathbf{v}\|_2^2}{2} \leqslant 0\right\} \tag{87}$$

which after straightforward calculations leads to (79). $\qquad\square$

**Lemma 6.** *By the definitions above, there exists finite constants* $K, K_1, K_2$ *and* $L, L_1, L_2$, *only depending on the common distribution of the entries of* $\mathbf{a}$, *such that*

$$\sqrt{\mathcal{E}_{\nu,\mathbf{a}}(\gamma^4)} \leqslant L_1 + K_1\frac{\|\hat{\mathbf{v}}\|_2^2}{m} \tag{88}$$

*Moreover,*

$$\left|\mathcal{E}_{\nu,\mathbf{a}}(\tilde{\Delta}) - \frac{\frac{1}{2m}(1 + \frac{\|\hat{\mathbf{v}}\|_2^2}{m})}{1 + \frac{Tr(\mathbf{\Omega}^{-1})}{m}}\right| \leqslant \frac{1}{\epsilon m\sqrt{m}}\left(L + K\frac{\|\hat{\mathbf{v}}\|_2^2}{m}\right) \tag{89}$$

*and*

$$\Pr\left(\sum_k |\Delta\tilde{v}_k|^3 > \eta\right) \leqslant \frac{\sqrt{L_2 + K_2\frac{\|\hat{\mathbf{v}}\|_2^6}{m^3}}}{\eta\epsilon^3\sqrt{m}} \tag{90}$$

*Proof.* Recall that

$$\tilde{\Delta} = \Psi(\mathbf{R}, \mathbf{a}, \boldsymbol{\nu}, \nu, \mathbf{x}) - \Phi(\mathbf{R}, \boldsymbol{\nu}, \mathbf{x}) = \frac{\frac{1}{2m}\gamma^2}{1 + \mathbf{a}^T\boldsymbol{\Omega}^{-1}\mathbf{a}} \tag{91}$$

where due to the second property of $f$, we have that

$$\boldsymbol{\Omega} = \mathbf{R}^T\mathbf{R} + \mathbf{H} \succ \epsilon\mathbf{I} \rightarrow \boldsymbol{\Omega}^{-1} \prec \frac{1}{\epsilon}\mathbf{I} \tag{92}$$

Note that the function $r(x) = 1/(1+x)$ is $1-$Lipshitz, which means that

$$\left| \frac{\frac{1}{2m}\gamma^2}{1 + \mathbf{a}^T\boldsymbol{\Omega}^{-1}\mathbf{a}} - \frac{\frac{1}{2m}\gamma^2}{1 + \frac{\boldsymbol{\Omega}^{-1}}{m}} \right| \leqslant \frac{\gamma^2}{2m}\left| \mathbf{a}^T\boldsymbol{\Omega}^{-1}\mathbf{a} - \frac{\boldsymbol{\Omega}^{-1}}{m} \right| \tag{93}$$

Thus,

$$\left| \mathcal{E}_{\nu,\mathbf{a}}\left( \frac{\frac{1}{2m}\gamma^2}{1+\mathbf{a}^T\boldsymbol{\Omega}^{-1}\mathbf{a}} \right) - \mathcal{E}_{\nu,\mathbf{a}}\left( \frac{\frac{1}{2m}\gamma^2}{1+\frac{\boldsymbol{\Omega}^{-1}}{m}} \right) \right| \leqslant \mathcal{E}_{\nu,\mathbf{a}}\left( \frac{\gamma^2}{2m}\left| \mathbf{a}^T\boldsymbol{\Omega}^{-1}\mathbf{a} - \frac{\boldsymbol{\Omega}^{-1}}{m} \right| \right)$$

$$\leqslant \frac{1}{2m}\sqrt{\mathcal{E}_{\nu,\mathbf{a}}\left( \left| \mathbf{a}^T\boldsymbol{\Omega}^{-1}\mathbf{a} - \frac{\boldsymbol{\Omega}^{-1}}{m} \right|^2 \right)\mathcal{E}_{\nu,\mathbf{a}}\left( \gamma^4 \right)} \tag{94}$$

By Lemma 3, we get that

$$\left| \mathcal{E}_{\nu,\mathbf{a}}\left( \frac{\frac{1}{2m}\gamma^2}{1+\mathbf{a}^T\boldsymbol{\Omega}^{-1}\mathbf{a}} \right) - \mathcal{E}_{\nu,\mathbf{a}}\left( \frac{\frac{1}{2m}\gamma^2}{1+\frac{\boldsymbol{\Omega}^{-1}}{m}} \right) \right| \leqslant \frac{K_0}{\epsilon m \sqrt{m}}\sqrt{\mathcal{E}_{\nu,\mathbf{a}}\left( \gamma^4 \right)} \tag{95}$$

where $K_0$ is a finite constant, only depending on the distribution $a$. On the other hand,

$$\mathcal{E}_{\nu,\mathbf{a}}\left( \gamma^4 \right) = \mathcal{E}_{\nu,\mathbf{a}}\left( (\nu + \mathbf{a}^T\hat{\mathbf{v}})^4 \right) = \mathcal{E}_{\nu,\mathbf{a}}\left( g^4 \right) + 6\mathcal{E}_{\nu,\mathbf{a}}\left( \nu^2 \right)\mathcal{E}_{\nu,\mathbf{a}}\left( (\mathbf{a}^T\hat{\mathbf{v}})^2 \right) + \mathcal{E}_{\nu,\mathbf{a}}\left( (\mathbf{a}^T\hat{\mathbf{v}})^4 \right) \tag{96}$$

Using Lemma 3 and after straightforward calculations, it is easy to see that there exists finite constants $K_1$ and $L_1$, only depending on $a$ such that

$$\sqrt{\mathcal{E}_{\nu,\mathbf{a}}\left( \gamma^4 \right)} \leqslant L_1 + K_1\frac{\|\hat{\mathbf{v}}\|_2^2}{m} \tag{97}$$

Combining this with (95), we obtain that there exists constants $K = K_0K_1$ and $L = K_0L_1$ such that

$$\left| \mathcal{E}_{\nu,\mathbf{a}}\left( \frac{\frac{1}{2m}\gamma^2}{1+\mathbf{a}^T\boldsymbol{\Omega}^{-1}\mathbf{a}} \right) - \mathcal{E}_{\nu,\mathbf{a}}\left( \frac{\frac{1}{2m}\gamma^2}{1+\frac{\boldsymbol{\Omega}^{-1}}{m}} \right) \right| \leqslant \frac{1}{\epsilon m \sqrt{m}}\left( L + K\frac{\|\hat{\mathbf{v}}\|_2^2}{m} \right) \tag{98}$$

which can also be written as

$$\left| \mathcal{E}_{\nu,\mathbf{a}}(\tilde{\Delta}) - \frac{\frac{1}{2m}(1 + \frac{\|\hat{\mathbf{v}}\|_2^2}{m})}{1 + \frac{\boldsymbol{\Omega}^{-1}}{m}} \right| \leqslant \frac{1}{\epsilon m \sqrt{m}}\left( L + K\frac{\|\hat{\mathbf{v}}\|_2^2}{m} \right) \tag{99}$$

Finally, defining $\boldsymbol{\omega}_k$ as the $k^{\text{th}}$ colum of $\boldsymbol{\Omega}^{-1}$ we observe that $\|\boldsymbol{\omega}_k\|_2 \leqslant 1/\epsilon$, since $\boldsymbol{\Omega}^{-1} \prec \mathbf{I}/\epsilon$. Now, using Lemma 3,

$$\mathcal{E}_{\nu,\mathbf{a}}(\sum_k |\Delta\tilde{v}_k|^3) = \sum_k \mathcal{E}_{\nu,\mathbf{a}}(|\gamma|^3|\boldsymbol{\omega}_k^T\mathbf{a}|^3) \leqslant \sum_k \sqrt{\mathcal{E}_{\nu,\mathbf{a}}(\gamma^6)\mathcal{E}_{\nu,\mathbf{a}}(|\boldsymbol{\omega}_k^T\mathbf{a}|^6)} \leqslant \sqrt{\mathcal{E}_{\nu,\mathbf{a}}(\gamma^6)}\frac{1}{\epsilon^3\sqrt{m}} \tag{100}$$

On the other hand,

$$\sqrt{\mathcal{E}_{\nu,\mathbf{a}}(\gamma^6)} \leqslant \sqrt{32\mathcal{E}_{\nu,\mathbf{a}}(\nu^6 + |\hat{\mathbf{v}}^T\mathbf{a}|^6)} \leqslant \sqrt{L_2 + K_2\frac{\|\hat{\mathbf{v}}\|_2^6}{m^3}} \tag{101}$$

Using the Markov's inequality, we obtain that

$$\Pr\left( \sum_k |\Delta\tilde{v}_k|^3 > \eta \right) \leqslant \frac{\sqrt{L_2 + K_2\frac{\|\hat{\mathbf{v}}\|_2^6}{m^3}}}{\eta\epsilon^3\sqrt{m}} \tag{102}$$

$\square$

**Lemma 7.** *We have that*

$$|\Delta' - \tilde{\Delta}| \leqslant \frac{1}{2m}\gamma^2 \chi_{\{\|\tilde{\mathbf{v}}\|_3^3 > \eta_1\}} + \frac{\epsilon\epsilon_1}{m}\|\tilde{\mathbf{v}}\|_3^3 \chi_{\{\|\tilde{\mathbf{v}}\|_3^3 < \eta_1\}} \tag{103}$$

*Proof.* Denote by $\Psi(\mathbf{v}, \mathbf{R}, \mathbf{a}, \boldsymbol{\mu}, \nu, \mathbf{x})$ and $\Theta(\mathbf{v}, \mathbf{R}, \mathbf{a}, \boldsymbol{\mu}, \nu, \mathbf{x})$ the cost functions in (45) and (28). Note that $\Psi(\mathbf{R}, \mathbf{a}, \boldsymbol{\mu}, \nu, \mathbf{x})$ and $\Theta(\mathbf{R}, \mathbf{a}, \boldsymbol{\mu}, \nu, \mathbf{x})$ are the minimum values of $\Psi(\mathbf{v}, \mathbf{R}, \mathbf{a}, \boldsymbol{\mu}, \nu)$ and $\Theta(\mathbf{v}, \mathbf{R}, \mathbf{a}, \boldsymbol{\mu}, \nu)$ over $\mathbf{v}$. Applying the mean value theorem, we conclude that for any point $\mathbf{v}$, it holds that

$$\Psi(\mathbf{v}, \mathbf{R}, \mathbf{a}, \boldsymbol{\mu}, \nu) - \Theta(\mathbf{v}, \mathbf{R}, \mathbf{a}, \boldsymbol{\mu}, \nu) = \frac{1}{m}\sum_{\alpha,\beta,\gamma}\frac{\partial^3 f(\mathbf{v}' + \mathbf{x}) + \rho g(\mathbf{v}')}{\partial x_\alpha \partial x_\beta \partial x_\gamma}\Delta v_\alpha \Delta v_\beta \Delta v_\gamma \tag{104}$$

For a proper point $\mathbf{v}'$. Due to the third property of $f$ and $g$, we conclude that

$$|\Psi(\mathbf{v}, \mathbf{R}, \mathbf{a}, \boldsymbol{\mu}, \nu) - \Theta(\mathbf{v}, \mathbf{R}, \mathbf{a}, \boldsymbol{\mu}, \nu)| \leqslant \frac{\bar{C}_2}{m}\sum_\alpha |\Delta v_\alpha|^3 \tag{105}$$

where $\bar{C}_2 = C_2 + \rho\tilde{C}_2$ and we remind that $\Delta\mathbf{v} = \mathbf{v} - \hat{\mathbf{v}}$. On the other hand,

$$\Psi(\mathbf{v}, \mathbf{R}, \mathbf{a}, \boldsymbol{\mu}, \nu, \mathbf{x}) - \Theta(\mathbf{R}, \mathbf{a}, \boldsymbol{\mu}, \nu, \mathbf{x}) = \frac{1}{m}(\Delta\mathbf{v} - \Delta\tilde{\mathbf{v}})^T(\boldsymbol{\Omega} + \mathbf{a}\mathbf{a}^T)(\Delta\mathbf{v} - \Delta\tilde{\mathbf{v}}) \tag{106}$$

which yields to

$$\Psi(\mathbf{v}, \mathbf{R}, \mathbf{a}, \boldsymbol{\mu}, \nu, \mathbf{x}) - \Psi(\mathbf{R}, \mathbf{a}, \boldsymbol{\mu}, \nu, \mathbf{x}) \geqslant \frac{\epsilon}{m}\|\Delta\mathbf{v} - \Delta\tilde{\mathbf{v}}\|_2^2 \tag{107}$$

For $\alpha = \epsilon/C_2$, take $\eta_1, \epsilon_1$ as in Lemma 2. Now, we consider two cases:

**Case 1:** If $\|\Delta\tilde{\mathbf{v}}\|_3^3 \leqslant \eta_1$, one can choose $r \leqslant \epsilon_1\|\Delta\tilde{\mathbf{v}}\|_3^3$ such that

$$4r^{\frac{3}{2}} + 5\|\Delta\tilde{\mathbf{v}}\|_3^3 < \frac{\epsilon}{\bar{C}_2}r \tag{108}$$

Note that if $\|\Delta\mathbf{v} - \Delta\tilde{\mathbf{v}}\|_2^2 \leqslant r$, we have that

$$\frac{\bar{C}_2}{\epsilon}(\|\Delta\mathbf{v}\|_3^3 + \|\Delta\tilde{\mathbf{v}}\|_3^3) \leqslant \frac{\bar{C}_2}{\epsilon}(4\|\Delta\mathbf{v} - \Delta\tilde{\mathbf{v}}\|_3^3 + 4\|\Delta\tilde{\mathbf{v}}\|_3^3 + \|\Delta\tilde{\mathbf{v}}\|_3^3) \leqslant$$
$$\frac{\bar{C}_2}{\epsilon}(4r^{\frac{3}{2}} + 5\|\Delta\tilde{\mathbf{v}}\|_3^3) < r \tag{109}$$

Thus,

$$\|\Delta\mathbf{v} - \Delta\tilde{\mathbf{v}}\|_2^2 = r \to \Theta(\hat{\mathbf{v}} + \Delta\mathbf{v}, \mathbf{R}, \mathbf{a}, \boldsymbol{\mu}, \nu, \mathbf{x}) - \Theta(\hat{\mathbf{v}} + \Delta\tilde{\mathbf{v}}, \mathbf{R}, \mathbf{a}, \boldsymbol{\mu}, \nu, \mathbf{x}) \geqslant$$
$$\Psi(\hat{\mathbf{v}} + \Delta\mathbf{v}, \mathbf{R}, \mathbf{a}, \boldsymbol{\mu}, \nu, \mathbf{x}) - \Psi(\mathbf{R}, \mathbf{a}, \boldsymbol{\mu}, \nu, \mathbf{x}) - \frac{C_2}{m}\sum_\alpha |\Delta v_k|^3 + |\Delta\tilde{v}_k|^3 \geqslant$$
$$\frac{\epsilon}{m}\|\Delta\mathbf{v} - \Delta\tilde{\mathbf{v}}\|_2^2 - \frac{C_2}{m}\sum_\alpha |\Delta v_k|^3 + |\Delta\tilde{v}_k|^3 > 0 \tag{110}$$

This means that, $\Phi$ has a local minimum in the ball $B_r = \{\hat{\mathbf{v}} + \Delta\mathbf{v} \mid \|\Delta\mathbf{v} - \Delta\tilde{\mathbf{v}}\|_2^2 \leqslant r\}$ which due to convexity, is also the global minimal point. We also conclude that

$$\Theta(\mathbf{R}, \mathbf{a}, \boldsymbol{\mu}, \nu, \mathbf{x}) = \min_{\Delta\mathbf{v} \in B_r} \Theta(\hat{\mathbf{v}} + \Delta\mathbf{v}, \mathbf{R}, \mathbf{a}, \boldsymbol{\mu}, g) \tag{111}$$

But for any $\Delta\mathbf{v} \in B_r$,

$$|\Psi(\mathbf{v}, \mathbf{R}, \mathbf{a}, \boldsymbol{\mu}, \nu, \mathbf{x}) - \Phi(\mathbf{v}, \mathbf{R}, \mathbf{a}, \boldsymbol{\mu}, \nu, \mathbf{x})| \leqslant \frac{C_2}{m}\sum_\alpha |\Delta\tilde{v}_\alpha|^3 + |\Delta v_\alpha|^3 \leqslant \frac{\epsilon r}{m} \tag{112}$$

Thus,

$$\|\Delta\tilde{\mathbf{v}}\|_3^3 \leqslant \eta_1 \to |\Psi(\mathbf{R}, \mathbf{a}, \boldsymbol{\mu}, \nu, \mathbf{x}) - \Phi(\mathbf{R}, \mathbf{a}, \boldsymbol{\mu}, \nu, \mathbf{x})| \leqslant \frac{\epsilon r}{m} \tag{113}$$

which, recalling that $\Delta' = \Theta(\mathbf{R}, \mathbf{a}, \boldsymbol{\mu}, \nu, \mathbf{x}) - \Phi(\mathbf{R}, \boldsymbol{\mu}, \nu, \mathbf{x})$ and $\tilde{\Delta} = \Psi(\mathbf{R}, \mathbf{a}, \boldsymbol{\mu}, \nu, \mathbf{x}) - \Phi(\mathbf{R}, \boldsymbol{\mu}, \mathbf{x})$, can be written as

$$\|\Delta\tilde{\mathbf{v}}\|_3^3 \leqslant \eta_1 \to |\Delta' - \tilde{\Delta}| \leqslant \frac{\epsilon r}{m} \leqslant \frac{\epsilon\epsilon_1}{m}\sum_\alpha |\Delta\tilde{v}_k|^3 \tag{114}$$

**Case 2:** If $\|\tilde{\mathbf{v}}\|_3^3 > \eta_1$, we may use the following simpler bound:

$$\Phi(\mathbf{R}, \boldsymbol{\mu}, \mathbf{x}) \leqslant \Theta(\mathbf{R}, \mathbf{a}, \boldsymbol{\mu}, \nu, \mathbf{x}) \leqslant \Theta(\hat{\mathbf{v}}, \mathbf{R}, \mathbf{a}, \boldsymbol{\mu}, \nu, \mathbf{x}) = \Phi(\mathbf{R}, \boldsymbol{\mu}, \mathbf{x}) + \frac{1}{2m}\gamma^2 \tag{115}$$

Thus,

$$-\frac{\frac{1}{2m}\gamma^2}{1 + \mathbf{a}^T\boldsymbol{\Omega}^{-1}\mathbf{a}} < \Delta' - \tilde{\Delta} \leqslant \frac{\frac{1}{2m}\gamma^2\mathbf{a}^T\boldsymbol{\Omega}^{-1}\mathbf{a}}{1 + \mathbf{a}^T\boldsymbol{\Omega}^{-1}\mathbf{a}} \tag{116}$$

which yields to

$$|\Delta' - \tilde{\Delta}| \leqslant \frac{1}{2m}\gamma^2 \tag{117}$$

This gives the result. $\qquad\square$

## 3.2 Proof of Theorem 3

### 3.2.1 Part 1

Note that Lemma 1 implies the first part of Theorem 3. To see this, set $\rho = 0$, and note that by the definition of a regular function $f_n$ there exists a family of smooth-regular sequences $f_n^{(k)}(\mathbf{x})$ such that $f_n^{(k)}(\mathbf{x})/n$ converges uniformly in $n, x$ to $f_n(x)/n$. Define $\Phi_{n,\rho}^{(k)}(\mathbf{A}, \boldsymbol{\nu}, \mathbf{x}_0)$ as the optimal cost in (1 in Paper), when $f_n$ is substituted by $f_n^{(k)}$. Now assume that $\Phi_n(\mathbf{A}_1, \boldsymbol{\nu}, \mathbf{x}_0) \to_p C$. Take $\delta > 0$ and note that

$$\lim_{n \to \infty} \Pr(|\Phi_n(\mathbf{A}_1, \boldsymbol{\nu}, \mathbf{x}_0) - C| > \delta) \to 0 \tag{118}$$

Take $k$ large enough such that $|f_n^{(k)}(\mathbf{x}) - f_n(\mathbf{x})| < n\delta$ holds for every $k, n$. Then, we have that $|\Phi_n^{(k)} - \Phi_n| < \delta$ and

$$\Pr(|\Phi_n^{(k)}(\mathbf{A}_1, \boldsymbol{\nu}, \mathbf{x}_0) - C| > 2\delta) \leqslant \Pr(|\Phi_n(\mathbf{A}_1, \boldsymbol{\nu}, \mathbf{x}_0) - C| > \delta) \tag{119}$$

letting $n \to \infty$, we get that

$$\lim_{n \to \infty} \Pr(|\Phi_n^{(k)}(\mathbf{A}_1, \boldsymbol{\nu}, \mathbf{x}_0) - C| > 2\delta) \to 0 \tag{120}$$

Now, from Lemma 1, we get that

$$\lim_{n \to \infty} \Pr(|\Phi_n^{(k)}(\mathbf{A}_2, \boldsymbol{\nu}, \mathbf{x}_0) - C| > 3\delta) \to 0 \tag{121}$$

On the other hand, from the fact that $|\Phi_n^{(k)} - \Phi_n| < \delta$, we get that

$$\Pr(|\Phi_n(\mathbf{A}_2, \boldsymbol{\nu}, \mathbf{x}_0) - C| > 4\delta) \leqslant \Pr(|\Phi_n^{(k)}(\mathbf{A}_1, \boldsymbol{\nu}, \mathbf{x}_0) - C| > 3\delta) \tag{122}$$

letting $n \to \infty$, we obtain that

$$\lim_{n \to \infty} \Pr(|\Phi_n(\mathbf{A}_2, \boldsymbol{\nu}, \mathbf{x}_0) - C| > 4\delta) \to 0 \tag{123}$$

Since $\delta$ is arbitrary, we conclude that $\Phi_n(\mathbf{A}_2, \boldsymbol{\nu}, \mathbf{x}_0) \to_p C$.

### 3.2.2 Part 2

Suppose that $0 \leqslant g'' < C$ and $f$ is $\epsilon$−strongly convex. Note that the functions $f_n(\mathbf{v} + \mathbf{x}_0) + \rho g(\mathbf{v})$ of $\mathbf{v}$ are convex for $\rho \geqslant -\epsilon/C$. Take $\rho \geqslant -\epsilon/C$. Define $X$ and $\Gamma$ as two independent random variables with $\xi$ and standard normal distributions, respectively. For a fixed value of $k$, take the following modified Key optimizations

$$\phi_{n,\rho}(\mathbf{g}, \mathbf{x}_0) = \max_{\beta > 0} \min_{\mathbf{v} \in \mathbb{R}^n} \frac{m\beta}{n}\sqrt{\sigma^2 + \frac{\|\mathbf{v}\|_2^2}{m}} + \beta\frac{\mathbf{g}^T\mathbf{v}}{n} - \frac{m}{2n}\beta^2 + \frac{f_n(\mathbf{v} + \mathbf{x}_0) + \rho g_n(\mathbf{v})}{n} \tag{124}$$

and

$$\phi'_{n,\rho}(\mathbf{g}, \mathbf{x}_0) = \max_{\beta > 0} \min_{\mathbf{v} \in \mathbb{R}^n} \frac{m\beta}{n}\sqrt{\sigma^2 + \frac{\|\mathbf{v}\|_2^2}{m}} + \beta\frac{\mathbf{g}^T\mathbf{v}}{n} - \frac{m}{2n}\beta^2 + \frac{f_n(\mathbf{v} + \mathbf{x}_0) + \rho g_n(\mathbf{v} + \mathbf{x}_0)}{n} \tag{125}$$

Furthermore, define the optimization

$$\Phi'_{n,\rho}(\mathbf{A}, \boldsymbol{\nu}, \mathbf{x}_0) = \frac{1}{n} \min_{\mathbf{v}} \frac{1}{2} \|\boldsymbol{\nu} + \mathbf{A}\mathbf{v}\|_2^2 + f_n(\mathbf{v} + \mathbf{x}_0) + \rho g_n(\mathbf{v} + \mathbf{x}_0) \tag{126}$$

where $f_n$ and $g_n$ are given by (1). In fact, (126) is in the same form as (18), where $f$ and $g$ are replaced by $f + \rho g$ and 0, respectively.

**step 1:** First take a Gaussian scaled-regular matrix $\mathbf{A}$. Similar to [2], applying the Gordon's Theorem to the primal and dual optimizations provides that if (124) converges to a value $C$, so does (18). A similar result hods for (125) and (126). In this case, denote the optimal values in (18) and (126) by $\Phi_{n,\rho}(\mathbf{A}, \boldsymbol{\nu}, \mathbf{x}_0)$ and $\Phi'_{n,\rho}(\mathbf{A}, \boldsymbol{\nu}, \mathbf{x}_0)$, respectively. Then by the method in [3], it is simple to see that

$$\phi_{n,\rho}(\mathbf{g}, \mathbf{x}_0) \to_p C_\rho = \max_{\beta} \min_{p} \left\{ \frac{p\beta\gamma}{2} + \frac{\gamma\beta\sigma^2}{2p} - \frac{\gamma\beta^2}{2} + \bar{H}_\rho(\beta, p) \right\} \tag{127}$$

and

$$\phi'_{n,\rho}(\mathbf{g}, \mathbf{x}_0) \to_p C'_\rho = \max_{\beta} \min_{p} \left\{ \frac{p\beta\gamma}{2} + \frac{\gamma\beta\sigma^2}{2p} - \frac{\gamma\beta^2}{2} + \bar{H}'_\rho(\beta, p) \right\} \tag{128}$$

where

$$\bar{H}_\rho(\beta, p) = \mathcal{E}_{\Gamma, X} \left[ \min_{v \in \mathbb{R}} \frac{\gamma^2 \beta}{2p} v^2 + \beta \Gamma v + f(v + X) + \rho g(v) \right] \tag{129}$$

and

$$\bar{H}'_\rho(\beta, p) = \mathcal{E}_{\Gamma, X} \left[ \min_{v \in \mathbb{R}} \frac{\gamma^2 \beta}{2p} v^2 + \beta \Gamma v + f(v + X) + \rho g(v + X) \right] \tag{130}$$

This guarantees that $\Phi_{n,\rho}(\mathbf{A}, \boldsymbol{\nu}, \mathbf{x}_0) \to_p C_\rho$ and $\Phi'_{n,\rho}(\mathbf{A}, \boldsymbol{\nu}, \mathbf{x}_0) \to_p C'_\rho$ for $\rho > -\epsilon/C$. According to Lemma 1, $\Phi_{n,\rho}(\mathbf{A}, \boldsymbol{\nu}, \mathbf{x}_0) \to_p C_\rho$ and $\Phi'_{n,\rho}(\mathbf{A}, \boldsymbol{\nu}, \mathbf{x}_0) \to_p C'_\rho$ holds for $\rho > -\epsilon/C$ and any scaled regular matrix $\mathbf{A}$.

**Step 2:** Note that $\Phi_{n,\rho}(\mathbf{A}, \boldsymbol{\nu}, \mathbf{x}_0)$ and $\Phi'_{n,\rho}(\mathbf{A}, \boldsymbol{\nu}, \mathbf{x}_0)$ are concave functions of $\rho$. Further,

$$\frac{g_n(\mathbf{w}(\mathbf{A}, \boldsymbol{\nu}, \mathbf{x}_0))}{n} = \left. \frac{\partial \Phi_{n,\rho}(\mathbf{A}, \boldsymbol{\nu}, \mathbf{x}_0)}{\partial \rho} \right|_{\rho=0} \tag{131}$$

and

$$\frac{g_n(\hat{\mathbf{x}}(\mathbf{A}, \boldsymbol{\nu}, \mathbf{x}_0))}{n} = \left. \frac{\partial \Phi'_{n,\rho}(\mathbf{A}, \boldsymbol{\nu}, \mathbf{x}_0)}{\partial \rho} \right|_{\rho=0} \tag{132}$$

Hence,

$$\frac{\Phi_{n,\rho}(\mathbf{A}, \boldsymbol{\nu}, \mathbf{x}_0) - \Phi_{n,0}(\mathbf{A}, \boldsymbol{\nu}, \mathbf{x}_0)}{\rho} \leqslant \frac{g_n(\mathbf{w}(\mathbf{A}, \boldsymbol{\nu}, \mathbf{x}_0))}{n} \leqslant \frac{\Phi_{n,0}(\mathbf{A}, \boldsymbol{\nu}, \mathbf{x}_0) - \Phi_{n,-\rho}(\mathbf{A}, \boldsymbol{\nu}, \mathbf{x}_0)}{\rho} \tag{133}$$

and

$$\frac{\Phi'_{n,\rho}(\mathbf{A}, \boldsymbol{\nu}, \mathbf{x}_0) - \Phi'_{n,0}(\mathbf{A}, \boldsymbol{\nu}, \mathbf{x}_0)}{\rho} \leqslant \frac{g_n(\hat{\mathbf{x}}(\mathbf{A}, \boldsymbol{\nu}, \mathbf{x}_0))}{n} \leqslant \frac{\Phi'_{n,0}(\mathbf{A}, \boldsymbol{\nu}, \mathbf{x}_0) - \Phi'_{n,-\rho}(\mathbf{A}, \boldsymbol{\nu}, \mathbf{x}_0)}{\rho} \tag{134}$$

Thus, for sufficiently small values of $\rho, \delta > 0$, we have that

$$\Pr\left( \frac{g_n(\mathbf{w}(\mathbf{A}, \boldsymbol{\nu}, \mathbf{x}_0))}{n} < \frac{C_\rho - C_0}{\rho} - \frac{\delta}{2} \right) \to 0 \tag{135}$$

and

$$\Pr\left( \frac{g_n(\mathbf{w}(\mathbf{A}, \boldsymbol{\nu}, \mathbf{x}_0))}{n} > \frac{C_0 - C_{-\rho}}{\rho} + \frac{\delta}{2} \right) \to 0 \tag{136}$$

Also,

$$\Pr\left( \frac{g_n(\hat{\mathbf{x}}(\mathbf{A}, \boldsymbol{\nu}, \mathbf{x}_0))}{n} < \frac{C'_\rho - C'_0}{\rho} - \frac{\delta}{2} \right) \to 0 \tag{137}$$

and

$$\Pr\left( \frac{g_n(\hat{\mathbf{x}}(\mathbf{A}, \boldsymbol{\nu}, \mathbf{x}_0))}{n} > \frac{C'_0 - C'_{-\rho}}{\rho} + \frac{\delta}{2} \right) \to 0 \tag{138}$$

Note that for the values of $\rho$ with sufficiently small absolute value, we have that

$$\left| \frac{C_0 - C_\rho}{\rho} - \frac{\partial C_\rho}{\partial \rho}\bigg|_{\rho=0} \right| \leqslant \frac{\delta}{2} \tag{139}$$

and

$$\left| \frac{C_0' - C_\rho'}{\rho} - \frac{\partial C_\rho}{\partial \rho}\bigg|_{\rho=0} \right| \leqslant \frac{\delta}{2} \tag{140}$$

The uniqueness of the values $\hat{p}$ and $\hat{\beta}$ guarantees that the derivatives exist. In this case,

$$\Pr\left( \left| \frac{g_n(\mathbf{w}(\mathbf{A}, \boldsymbol{\nu}, \mathbf{x}_0))}{n} - \frac{\partial C_\rho}{\partial \rho}\bigg|_{\rho=0} \right| > \delta \right) \to 0 \tag{141}$$

and

$$\Pr\left( \left| \frac{g_n(\hat{\mathbf{x}}(\mathbf{A}, \boldsymbol{\nu}, \mathbf{x}_0))}{n} - \frac{\partial C_\rho'}{\partial \rho}\bigg|_{\rho=0} \right| > \delta \right) \to 0 \tag{142}$$

which yield to

$$\frac{g_n(\mathbf{w}(\mathbf{A}, \boldsymbol{\nu}, \mathbf{x}_0))}{n} \to_p \frac{\partial C_\rho}{\partial \rho}\bigg|_{\rho=0} \tag{143}$$

and

$$\frac{g_n(\hat{\mathbf{x}}(\mathbf{A}, \boldsymbol{\nu}, \mathbf{x}_0))}{n} \to_p \frac{\partial C_\rho'}{\partial \rho}\bigg|_{\rho=0} \tag{144}$$

Finally, simple calculations show that if the solutions of (2) $\hat{p} = \hat{p}(\gamma, \sigma), \hat{\beta} = \hat{\beta}(\gamma, \sigma)$ is unique, then

$$\frac{\partial C_\rho}{\partial \rho}\bigg|_{\rho=0} = \mathcal{E}\left( g\left( \hat{x}_f(\frac{\gamma^2 \hat{\beta}}{2p}, \hat{\beta}\Gamma - \frac{\hat{\beta}\gamma^2 X}{\hat{p}}) - X \right) \right) \tag{145}$$

and

$$\frac{\partial C_\rho'}{\partial \rho}\bigg|_{\rho=0} = \mathcal{E}\left( g\left( \hat{x}_f(\frac{\gamma^2 \hat{\beta}}{2p}, \hat{\beta}\Gamma - \frac{\hat{\beta}\gamma^2 X}{\hat{p}}) \right) \right) \tag{146}$$

This proves part two of Theorem 1.

**Step 3:** For the third part in Theorem 1, define $g(x) = g_{0,x}(v) = \chi_{[x,\infty)}(v)$ and $g_{1,x}(\nu) = (\nu - x)_+$. Notice that there exists a sequence of convex functions with bounded second derivatives uniformly converging to $g_{1,x}$. Since, part two of Theorem 1 holds for any function in this sequence, it also holds for $g_{1,x}$. Now, take $g_{2,x,\epsilon} = (g_{1,x} - g_{1,x+\epsilon})/\epsilon$. Note that

$$g_{2,x,\epsilon} \leqslant g_{0,x} \leqslant g_{2,x-\epsilon,\epsilon} \tag{147}$$

For any $\delta > 0$, take $\epsilon > 0$ such that $L_{f,\chi_{[x,\infty)}}(\gamma, \sigma) - L_{f,\chi_{[x-\epsilon,\infty)}}(\gamma, \sigma) > -\delta/2$ and $M_{f,\chi_{[x,\infty)}}(\gamma, \sigma) - M_{f,\chi_{[x-\epsilon,\infty)}}(\gamma, \sigma) > -\delta/2$. Then,

$$\frac{g_n(\mathbf{w})}{n} - L_{f,\chi_{[x,\infty)}}(\gamma, \sigma) > \delta \to \frac{\sum_i g_{2,x-\epsilon,\epsilon}(w_i)}{n} - L_{f,\chi_{[x,\infty)}}(\gamma, \sigma) > \delta$$

$$\to \frac{\sum_i g_{2,x-\epsilon,\epsilon}(w_i)}{n} - L_{f,g_{2,x-\epsilon,\epsilon}}(\gamma, \sigma) > \delta + L_{f,\chi_{[x,\infty)}}(\gamma, \sigma) - L_{f,g_{2,x-\epsilon,\epsilon}}(\gamma, \sigma) > \delta + L_{f,\chi_{[x,\infty)}}(\gamma, \sigma) - L_{f,\chi_{[x-\epsilon,\infty)}}(\gamma, \sigma)$$

$$> \frac{\delta}{2} \tag{148}$$

With a similar approach,

$$\frac{g_n(\hat{\mathbf{x}})}{n} - M_{f,\chi_{[x,\infty)}}(\gamma, \sigma) > \delta \to \frac{\sum_i g_{2,x-\epsilon,\epsilon}(\hat{x}_i)}{n} - M_{f,g_{2,x-\epsilon,\epsilon}}(\gamma, \sigma) > \frac{\delta}{2} \tag{149}$$

The above results show that

$$\Pr\left( \frac{g_n(\mathbf{w})}{n} - L_{f,\chi_{[x,\infty)}}(\gamma, \sigma) > \delta \right) \leqslant \Pr\left( \frac{\sum_i g_{2,x-\epsilon,\epsilon}(w_i)}{n} - L_{f,g_{2,x-\epsilon,\epsilon}}(\gamma, \sigma) > \frac{\delta}{2} \right) \to 0 \tag{150}$$

and

$$\Pr\left(\frac{g_n(\hat{\mathbf{x}})}{n} - M_{f,\chi_{[x,\infty)}}(\gamma,\sigma) > \delta\right) \leqslant \Pr\left(\frac{\sum_i g_{2,x-\epsilon,\epsilon}(\hat{x}_i)}{n} - M_{f,g_{2,x-\epsilon,\epsilon}}(\gamma,\sigma) > \frac{\delta}{2}\right) \to 0$$

(151)

The other side of inequalities can be similarly shown, which yields to the desired result.

### 3.3 Proof of Theorem 2

#### 3.3.1 Part 1

Take function $f_\epsilon(x) = \lambda|x| + \frac{\epsilon}{2}x^2$ and notice that for any value of $\epsilon > 0$, $f_\epsilon$ satisfies the conditions of theorem 1. The idea is to show that removing the term $\frac{\epsilon}{2}x^2$ for a small value of $\epsilon$ may not dramatically change the optimal value. For that, we first introduce the following definition:

**Definition 5.** *Consider a $m \times n$ matrix $\mathbf{A}$. We define $\theta_k(\mathbf{A})$ for any $k < n/2$ as the smallest number $\theta$, such that for any disjoint index subsets $I, I' \subset \{1, 2, \ldots, n\}$ with $|I|, |I'| \leqslant k$,*

$$\sigma_{max}\left(\mathbf{A}_{I'}^T \mathbf{A}_I\right) \leqslant \theta$$

(152)

It is well known that $\theta_k \leqslant \delta_{2k}$. Furthermore, we have the following result:

**Lemma 8.** *Suppose that the $m \times n$ matrix $\mathbf{A}$ is generated by a sub-Gaussian unit-variance random variable and $m, n$ grow to infinity such that $m/n \to \gamma > 0$. Then, there exist constants $\alpha, \beta, \epsilon > 0$, such that*

$$\lim_{n\to\infty} \Pr(\delta_{\alpha n}(\mathbf{A}) + \theta_{\alpha n}(\mathbf{A}) > 1 - \epsilon) = 0$$

(153)

$$\lim_{n\to\infty} \Pr(\sigma_{max}(\mathbf{A}) > \beta) = 0$$

(154)

*Proof.* Our proof is inspired by the method in []. We assume that $A$ is $\sigma^2-$subgaussian.

**Step 1** First, take a vector $\mathbf{x} \in S_n$, where $S_n$ is the surface of the unit sphere in $\mathbb{R}^n$. Note that $\mathbf{y} = \mathbf{A}\mathbf{x}$ is an i.i.d vector with $\sigma^2-$subgaussian entries. We get that

$$\forall r > 0, \ \Pr(Y > r) \leqslant \min_{\lambda>0} \mathcal{E}(e^{\lambda Y}) e^{-\lambda r} \leqslant \min_{\lambda>0} e^{\frac{\sigma^2\lambda^2}{2} - \lambda r} = e^{\frac{-r^2}{2\sigma^2}}$$

(155)

where $Y = \sqrt{m}y_1$ and $y_1$ is the first element of $\mathbf{y}$. Note that $\mathcal{E}(Y) = 0$ and $\mathcal{E}(Y^2) = 1$. Applying the same bound on $-Y$ gives that

$$\forall r > 0, \ \Pr(|Y| > r) \leqslant 2e^{\frac{-r^2}{2\sigma^2}}$$

(156)

Furthermore, using Tonelli's theorem we obtain that

$$\forall 0 < \lambda < \frac{1}{2\sigma^2}, \ \mathcal{E}(e^{\lambda Y^2}) = 1 + 2\lambda \int_0^\infty te^{\lambda t^2} \Pr(|Y| > t)dt \leqslant 1 + 2\lambda \int_0^\infty te^{(\lambda - \frac{1}{2\sigma^2})t^2} dt = 1 + \frac{2\lambda}{\frac{1}{2\sigma^2} - \lambda} = \frac{\frac{1}{2\sigma^2} + \lambda}{\frac{1}{2\sigma^2} - \lambda}$$

(157)

Then,

$$\Pr\left(\|\mathbf{y}\|_2^2 \geqslant \beta\right) = \Pr\left(\sum_{k=1}^m Y_k^2 \geqslant m\beta\right) \leqslant \min_{\lambda>0}\left(\mathcal{E}(e^{\lambda Y^2})e^{-\lambda\beta}\right)^m = H(\beta)^m \leqslant \min_{0<\lambda<1/2\sigma^2}\left(\frac{\frac{1}{2\sigma^2} + \lambda}{\frac{1}{2\sigma^2} - \lambda}e^{-\lambda\beta}\right)^m = K(\beta)^m$$

(158)

where $\{Y_k = \sqrt{m}y_k\}$ are i.i.d with the same distribution as $Y$ and

$$K(\beta) = \min_{0<\lambda<1/2} \frac{\frac{1}{2\sigma^2} + \lambda}{\frac{1}{2\sigma^2} - \lambda}e^{-\lambda\beta}$$

(159)

and

$$H(\beta) = \min_{\lambda>0} \mathcal{E}(e^{\lambda Y^2})e^{-\lambda\beta}$$

(160)

Note that $K(\beta) < 1$ for sufficiently large values of $\beta$. Moreover the cost in (160) at $\lambda = 0$ is 1 and has negative derivative if $\beta > \mathcal{E}(Y^2) = 1$, where $H(\beta) < 1$.

On the other hand

$$\mathrm{Pr}\left(\|\mathbf{y}\|_2^2 \leqslant \rho\right) = \mathrm{Pr}\left(\sum_{k=1}^{m} Y_k^2 \leqslant m\rho\right) \leqslant \min_{\lambda > 0}\left(\mathcal{E}(e^{-\lambda Y^2})e^{\lambda \beta}\right)^m = L(\rho)^m \qquad (161)$$

where

$$L(\rho) = \min_{\lambda > 0}\mathcal{E}(e^{-\lambda Y^2})e^{\lambda \rho} \qquad (162)$$

Note that at $\lambda = 0$, the cost in (162) is 1 and has negative derivative if $\rho < \mathcal{E}(Y^2) = 1$, in which case $L(\rho) < 1$.

**Step 2** With a simple volume packing argument, for every $\delta > 0$ and $n = 1, 2, \ldots$, there exists a set $G_n \subset S_n$ of maximally $\left(\frac{3}{\delta}\right)^n$ points such that for any $\mathbf{x} \in S_n$, there exists a pont $\mathbf{x}_1 \in G_n$ $\|\mathbf{x} - \mathbf{x}_1\|_2 < \delta$. Denote $B = \max_{\mathbf{x} \in G_n}\|\mathbf{A}\mathbf{x}\|_2$ and $A = \sigma_{max}(\mathbf{A}) = \max_{\mathbf{x} \in S_n}\|\mathbf{A}\mathbf{x}\|_2$ with the maximum at $\mathbf{x}_0$. Thus,

$$A = \|\mathbf{A}\mathbf{x}_0\|_2 \leqslant \|\mathbf{A}\mathbf{x}_1\|_2 + \|\mathbf{A}(\mathbf{x}_0 - \mathbf{x}_1)\|_2 \leqslant B + \delta A \qquad (163)$$

where $\mathbf{x}_1$ is the closest point in $G_n$ to $\mathbf{x}_0$. If $\delta < 1$ we obtain that

$$\sigma_{max}(\mathbf{A}) \leqslant \frac{\max\limits_{\mathbf{x} \in G_n}\|\mathbf{A}\mathbf{x}\|_2}{1 - \delta} \qquad (164)$$

repeating the same argument for the minimum singular value gives that

$$\sigma_{min}(\mathbf{A}) \geqslant \max_{\mathbf{x} \in G_n}\|\mathbf{A}\mathbf{x}\|_2 - \sigma_{max}(\mathbf{A})\delta \qquad (165)$$

**Step 3** Now, it is clear from (164) that

$$\mathrm{Pr}(\sigma_{max}(\mathbf{A}) > \beta) \leqslant \mathrm{Pr}(\max_{\mathbf{x} \in G_n}\|\mathbf{A}\mathbf{x}\|_2 > \beta(1 - \delta)) \leqslant K(\beta(1 - \delta))^m (\frac{3}{\delta})^n \qquad (166)$$

Fix $\delta < 1$ and note that $K(\beta) \to 0$ as $\beta \to \infty$. Thus, one can select $\beta$ large enough such that the right hand side tends to zero. This proves the second part.

Fix a value of $\epsilon < 1$ and take $\beta = 1 + \epsilon$ and $\rho = 1 - \epsilon$. Take any submatrix $\mathbf{A}_I$ of $\mathbf{A}$ with $|I| = k$ and note that the previous results also hold for $\mathbf{A}_I$. This means that

$$\mathrm{Pr}(\sigma_{max}(\mathbf{A}_I) > 1 + \epsilon) \leqslant H((1 + \epsilon)(1 - \delta))^m (\frac{3}{\delta})^k \qquad (167)$$

and

$$\mathrm{Pr}(\sigma_{min}(\mathbf{A}_I) < 1 - \epsilon) \leqslant \mathrm{Pr}\left(\max_{\mathbf{x} \in G_n}\|\mathbf{A}\mathbf{x}\|_2 - \sigma_{max}(\mathbf{A})\delta < 1 - \epsilon\right) \leqslant L(1 - \epsilon + (1 + \epsilon)\delta)^m (\frac{3}{\delta})^k + H((1 + \epsilon)(1 - \delta))^m (\frac{3}{\delta})^k \qquad (168)$$

Take $k = 2\alpha n$ and note that there are $\binom{n}{k} = O(\rho^n)$ combinations of $|I| = k$, where $\rho = \exp(1 - 2\alpha\log(2\alpha) - (1 - 2\alpha)\log(1 - 2\alpha))$. We finally, obtain that

$$\mathrm{Pr}(\delta_{2\alpha n} > \epsilon) = O\left((L(1 - \epsilon + (1 + \epsilon)\delta)^{\gamma n} + H((1 + \epsilon)(1 - \delta))^{\gamma n})(\frac{3}{\delta})^{2n\alpha}\rho^n\right) \qquad (169)$$

Fix $\delta < \epsilon/(1 + \epsilon)$, which guarantees that $L = L(1 - \epsilon + (1 + \epsilon)\delta) < 1$ and $H = H((1 + \epsilon)(1 - \delta)) < 1$. Then note that $(\frac{3}{\delta})^{2\alpha}\rho(\alpha) \to 1$ as $\alpha \to 0$. This means that we can select $\alpha$ small enough such that $H \times (\frac{3}{\delta})^{2\alpha}\rho(\alpha) < 1$ and $L \times (\frac{3}{\delta})^{2\alpha}\rho(\alpha) < 1$. For this value of $\alpha$, we get that

$$\mathrm{Pr}(\delta_{2\alpha n} > \epsilon) \qquad (170)$$

The first result is obtained by noting that $\delta_{\alpha n} + \theta_{\alpha n} \leqslant 2\delta_{2\alpha n}$. $\qquad \square$

Now, we show the following theorem. Then, from the above lemma the first part of Theorem 2 follows immediately.

**Theorem 4.** *Consider the conditions in the first part of Theorem 2 and $f(x) = \lambda|x|$. Denote,*

$$\Phi_\lambda = \frac{1}{n}\min_{\mathbf{v}} \frac{1}{2}\|\boldsymbol{\nu} + \mathbf{A}\mathbf{v}\|_2^2 + \lambda\|\mathbf{v} + \mathbf{x}\|_1 \tag{171}$$

*Further, assume that there exist constants $\alpha, \beta, \epsilon > 0$, such that*

$$\lim_{n\to\infty} \Pr(\delta_{\alpha n}(\mathbf{A}) + \theta_{\alpha n}(\mathbf{A}) > 1 - \epsilon) = 0 \tag{172}$$

$$\lim_{n\to\infty} \Pr(\sigma_{max}(\mathbf{A}) > \beta) = 0 \tag{173}$$

*Then,*

$$\Phi_\lambda \to_p C_{f=\lambda|x|}(\gamma, \sigma) \tag{174}$$

*Proof.* Take $\hat{\mathbf{v}}^{(0)}$ as the minimal point of the optimization

$$\Phi_{\lambda,\mu} = \frac{1}{m}\min_{\mathbf{v}} \frac{1}{2}\|\boldsymbol{\nu} + \mathbf{A}\mathbf{v}\|_2^2 + \lambda\|\mathbf{v} + \mathbf{x}\|_1 + \frac{\mu}{2}\|\mathbf{v}\|_2^2 \tag{175}$$

From Theorem 2, there exists a real number $L$, such that for every $\mu < 1$, $\|\hat{\mathbf{v}}^{(0)}\|_2^2/m < L^2$ with high probability (i.e. $\Pr(\|\hat{\mathbf{v}}^{(0)}\|_2^2/m \geqslant L^2) \to 0$ as dimensions grow). Define

$$\phi(\mathbf{v}) = \frac{1}{2}\|\boldsymbol{\nu} + \mathbf{A}\mathbf{v}\|_2^2 + \lambda\|\mathbf{v} + \mathbf{x}\|_1 \tag{176}$$

The KKT condition, implies that

$$-\mu\mathbf{v}^{(0)} \in \partial\phi(\mathbf{v}^{(0)}) \tag{177}$$

Define $\boldsymbol{\zeta}^{(0)} = -\mu\mathbf{v}^{(0)}$. Set $k = \alpha n$, select $k$ entries of $\mathbf{v}^{(0)}$ with largest absolute values and collect their indexes in $I_0$. Set $\mathbf{p}_0 = \mathbf{0} \in \mathbb{R}^k$ and $t = 0$. Now, perform the following iterative algorithm.

1. Define $\mathbf{P}_t = \mathbf{A}_{I_t}$ and $\mathbf{h}_t = \boldsymbol{\nu} + \mathbf{A}_{I_t^c}\mathbf{v}_{I_t^c}^{(t)}$, and solve

$$\min_{\mathbf{w}} \frac{1}{2}\|\mathbf{h}_t + \mathbf{P}_t\mathbf{w}\|_2^2 + \lambda\|\mathbf{x}_{I_t} + \mathbf{w}\| - \mathbf{p}_t^T\mathbf{w} \tag{178}$$

   Denote its cost function and minimum by $\phi_t(\mathbf{w})$ and $\mathbf{w}_t$, respectively.

2. Find $k$ elements in $I_t^c$ with largest absolute value in $\mathbf{A}_{I_t^c}^T\mathbf{A}_{I_t}(\mathbf{w}_t - \mathbf{v}_{I_t}^{(t)})$. Denote their indexes by $I_{t+1}$. Set $\mathbf{p}_{t+1} = \boldsymbol{\zeta}_{I_{t+1}}^{(t)}$.

3. Construct $\mathbf{v}^{(t+1)}$ and $\boldsymbol{\zeta}_{t+1}$, such that $\mathbf{v}_{I_t}^{(t+1)} = \mathbf{w}_t$, $\mathbf{v}_{I_t^c}^{(t+1)} = \mathbf{v}_{I_t^c}^{(t)}$, $\boldsymbol{\zeta}_{I_t}^{(t+1)} = \mathbf{p}_t$, and $\boldsymbol{\zeta}_{I_t^c}^{(t+1)} = \boldsymbol{\zeta}_{I_t^c}^{(t)} + \mathbf{A}_{I_t^c}^T\mathbf{A}_I(\mathbf{w}_t - \mathbf{v}_I^{(t)})$.

4. Set $t \leftarrow t + 1$ and go to step 1.

In the sequel, we show that the above process leads to a point $\mathbf{v}^{(\infty)}$ with a sub-gradient $\boldsymbol{\zeta}^{(\infty)} \in \partial\phi(\mathbf{v}^{(\infty)})$. Such that,

$$\frac{1}{\sqrt{m}}\|\mathbf{v}^{(\infty)} - \mathbf{v}^{(0)}\|_2 \leqslant \frac{\mu L}{1 - \delta_k - \theta_k} \tag{179}$$

$$\|\boldsymbol{\zeta}^{(\infty)}\|_\infty \leqslant \mu L\left(\sqrt{\frac{m}{k}} + \frac{\theta_k}{1 - \delta_k - \theta_k}\right) \tag{180}$$

We denote $C_1 = L\left(\sqrt{\frac{m}{k}} + \frac{\theta_k}{1 - \delta_k - \theta_k}\right)$ and $C_2 = \frac{L}{1 - \delta_k - \theta_k}$.

Once this is established, notice that $\mathbf{v}^{(\infty)}$ is the minimum point of the optimization

$$\rho_{\mu,\lambda} = \min_{\mathbf{v}} \frac{1}{2}\|\boldsymbol{\nu} + \mathbf{A}\mathbf{v}\|_2^2 + \lambda\|\mathbf{v} + \mathbf{x}\|_1 + \mathbf{v}^T\boldsymbol{\zeta}^{(\infty)} \tag{181}$$

The subscripts $\lambda, \mu$ emphasize that $\boldsymbol{\zeta}^{(\infty)}$, $\mathbf{v}^{(\infty)}$ are computed for a given $\lambda, \mu$. Note that since $\mathbf{v}^T\boldsymbol{\zeta}^{(\infty)} \leqslant \|\mathbf{v}\|_1\|\boldsymbol{\zeta}^{(\infty)}\|_\infty \leqslant \mu C_1\|\mathbf{v}\|_1$, we get that

$$\rho_{\mu,\lambda} \leqslant \Phi_{\lambda + C_1\mu} \tag{182}$$

which can also be written as

$$\Phi_\lambda \geqslant \rho_{\mu, \lambda - C_1 \mu} \tag{183}$$

On the other hand,

$$
\begin{aligned}
m\rho_{\mu,\lambda} &= \tfrac{1}{2}\|\boldsymbol{\nu} + \mathbf{A}\mathbf{v}^{(\infty)}\|_2^2 + \lambda\|\mathbf{v}^{(\infty)} + \mathbf{x}\|_1 + (\mathbf{v}^{(\infty)})^T\boldsymbol{\zeta}^{(\infty)} \\
&\geqslant m\Phi_{\lambda,\mu} + \mathbf{f}^T\mathbf{A}(\mathbf{v}^{(\infty)} - \mathbf{v}^{(0)}) - \lambda\|\mathbf{v}^{(\infty)} - \mathbf{v}^{(0)}\|_1 + (\mathbf{v}^{(\infty)})^T\boldsymbol{\zeta}^{(\infty)} \\
&\geqslant m\Phi_{\lambda,\mu} - \left(\|\mathbf{f}^T\mathbf{A}\|_2 + \lambda\sqrt{m}\right)\|\mathbf{v}^{(\infty)} - \mathbf{v}^{(0)}\|_2 - \|\mathbf{v}^{(\infty)}\|_2\|\boldsymbol{\zeta}^{(\infty)}\|_2 \\
&\quad m\Phi_{\lambda,\mu} - (\sigma_{\max}(\mathbf{A})\|\mathbf{f}\|_2 + \lambda\sqrt{m})\,\mu C_2\sqrt{m} - \sqrt{m}\mu C_1\|\mathbf{v}^{(\infty)}\|_2 \\
&\geqslant m\Phi_{\lambda,\mu} - (\sigma_{\max}(\mathbf{A})r + \lambda)\,\mu C_2 m - m\mu C_1(L + C_2\mu)
\end{aligned}
\tag{184}
$$

where $\mathbf{f} = \boldsymbol{\nu} + \mathbf{A}\mathbf{v}^{(0)}$, $r$ is a proper bound, independent of all parameters, such that $\|\boldsymbol{\nu}\|_2 \leqslant r\sqrt{m}$ with high probability (which exists by the law of large numbers) and we use the fact that $\|\mathbf{f}\|_2 \leqslant \|\boldsymbol{\nu}\|_2$. Thus,

$$\rho_{\mu,\lambda} \geqslant \Phi_{\lambda,\mu} - (\sigma_{\max}(\mathbf{A})r + \lambda)\,\mu C_2 - \mu C_1(L + C_2\mu) \tag{185}$$

We conclude that

$$\Phi_{\lambda,\mu} \geqslant \Phi_\lambda \geqslant \Phi_{\lambda - C_1\mu, \mu} - (\sigma_{\max}(\mathbf{A})r + \lambda - C_1\mu)\,\mu C_2 - \mu C_1(L + C_2\mu) \tag{186}$$

Noting that $\Phi_{\lambda,\mu} \to_p C_{\lambda,\mu} = C_{f = \lambda|x| + \mu/2|x|^2}$, given in Theorem 2, and due to continuity of $C_{\lambda,\mu}$ at $\mu = 0$, for any $\epsilon > 0$, one can select $\mu$ small enough such that $\Pr(|\Phi_\lambda - C_{\lambda,\mu=0}| > \epsilon) \to_p 0$. This completes the proof as $C_{\lambda,\mu=0} = C_{f=\lambda|x|}$.

Now, we show (179) and (180):

**Step 1:**

First note that $\boldsymbol{\zeta}_t \in \partial\phi(\mathbf{v}^{(n)})$. To see this, use induction:

- Clearly $\boldsymbol{\zeta}_0 \in \partial\phi(\mathbf{v}^{(0)})$.

- Suppose that $\boldsymbol{\zeta}_t \in \partial\phi(\mathbf{v}^{(t)})$. From the KKT condition for (178), we have that $(\boldsymbol{\zeta}_{t+1})_{I_t} = \mathbf{p}_t \in \mathbf{A}_{I_t}^H(g + \mathbf{A}\mathbf{v}^{(t+1)}) + \partial\|\mathbf{x}_{I_t} + \mathbf{v}_{I_t}^{(n+1)}\|_1$. Moreover, noting that $(\boldsymbol{\zeta}_t)_{I_t^c} \in \mathbf{A}_{I_t^c}^H(g + \mathbf{A}\mathbf{v}^{(t)}) + \partial\|\mathbf{x}_{I_t^c} + \mathbf{v}_{I_t^c}^{(t)}\|_1$, we get that $(\boldsymbol{\zeta}_{t+1})_{I_t^c} \in -\mathbf{A}_{I_t^c}^H(g + \mathbf{A}\mathbf{v}^{(t+1)}) + \partial\|\mathbf{x}_{I_t^c} + \mathbf{v}_{I_t^c}^{(t+1)}\|_1$. This shows that $\boldsymbol{\zeta}_{t+1} \in \partial\phi(\mathbf{v}^{(t+1)})$.

**Step 2:**

Now we show by induction that

$$\frac{1}{\sqrt{m}}\|\mathbf{v}^{(t+1)} - \mathbf{v}^{(t)}\|_2 \leqslant \frac{\mu L}{1 - \delta_k}\left(\frac{\theta_k}{1 - \delta_k}\right)^t \tag{187}$$

$$\boldsymbol{\zeta}_{I_t}^{(t+1)} = \boldsymbol{\zeta}_{I_t}^{(t-1)} \tag{188}$$

$$\|\boldsymbol{\zeta}_{(I_t \cup I_{t+1})^c}^{(t+1)} - \boldsymbol{\zeta}_{(I_t \cup I_{t+1})^c}^{(t)}\|_\infty \leqslant \mu L\left(\frac{\theta_k}{1 - \delta_k}\right)^{t+1}\sqrt{\frac{m}{k}} \tag{189}$$

The argument is as follows:

- First, note that (188) holds, since by definition, $\boldsymbol{\zeta}_{I_t}^{(t-1)} = \boldsymbol{\zeta}_{I_t}^{(t+1)} = \mathbf{p}_t$. Then,

$$\frac{1}{\sqrt{m}}\|\boldsymbol{\zeta}_{I_0}^{(0)}\|_2 \leqslant \frac{1}{\sqrt{m}}\|\boldsymbol{\zeta}^{(0)}\|_2 = \mu L \tag{190}$$

Thus, $\min|\boldsymbol{\zeta}_{I_0}^{(0)}| \leqslant \mu L\sqrt{\frac{m}{k}}$, which leads to

$$\|\boldsymbol{\zeta}_{I_0^c}^{(0)}\|_\infty \leqslant \min|\boldsymbol{\zeta}_{I_0}^{(0)}| \leqslant \mu L\sqrt{\frac{m}{k}} \tag{191}$$

Note that $\boldsymbol{\zeta}_{I_0}^{(0)} \in \partial\phi_0(\mathbf{w} = \mathbf{v}_{I_0}^{(0)})$. Hence, by Lemma 4, we get that

$$\frac{1}{\sqrt{m}}\|\mathbf{w}_0 - \mathbf{v}_{I_0}^{(0)}\|_2 \leqslant \frac{\|\boldsymbol{\zeta}_{I_0}^{(0)}\|_2}{\sigma_{\min}^2(\mathbf{A}_{I_0})} \leqslant \frac{\mu L}{1 - \delta_k} \tag{192}$$

- Start induction by $t = 0$: From the construction, we get

$$\begin{cases} \boldsymbol{\zeta}_{I_0}^{(1)} = \mathbf{0} \\ \boldsymbol{\zeta}_{I_0}^{(1)} = \boldsymbol{\zeta}_{I_0}^{(0)} + \mathbf{A}_{I_0^c}^T \mathbf{A}_{I_0}(\mathbf{w}_0 - \mathbf{v}_{I_0}^{(0)}) \end{cases} \tag{193}$$

and $\mathbf{p}_1 = \boldsymbol{\zeta}_{I_1}^{(0)}$. Note that

$$\frac{1}{\sqrt{m}}\left\|\mathbf{A}_{I_1}^T \mathbf{A}_{I_0}\left(\mathbf{w}_0 - \mathbf{v}_{I_0}^{(0)}\right)\right\|_2 \leqslant \frac{\theta_k}{\sqrt{m}}\left\|\mathbf{w}_0 - \mathbf{v}_{I_0}^{(0)}\right\|_2 \leqslant \frac{\theta_k \mu L}{1 - \delta_k} \tag{194}$$

which leads to

$$\left\|\mathbf{A}_{(I_0 \cup I_1)^c}^T \mathbf{A}_{I_0}\left(\mathbf{w}_0 - \mathbf{v}_{I_0}^{(0)}\right)\right\|_\infty \leqslant \min\left|\mathbf{A}_{I_1}^T \mathbf{A}_{I_0}\left(\mathbf{w}_0 - \mathbf{v}_{I_0}^{(0)}\right)\right| \leqslant \frac{\theta_k \mu L}{1 - \delta_k}\sqrt{\frac{m}{k}} \tag{195}$$

proving (189) for $t = 0$. Note that the relation in (192) proves (187) for $t = 0$ as $\|\mathbf{v}^{(1)} - \mathbf{v}^{(0)}\|_2 = \|\mathbf{w}_0 - \mathbf{v}_{I_0}^{(0)}\|_2$.

- Now, suppose that the relations (187),(188) and (189) hold for all $t' \leqslant t$ an let us prove them for $t + 1$. Consider the optimization (178) for $t$ and note that $\boldsymbol{\zeta}^{(t)} \in \partial\phi(\mathbf{v}^{(t)})$. It is simple to see that this leads to $\boldsymbol{\zeta}_{I_t}^{(t)} - \mathbf{p}_t \in \partial\phi_t(\mathbf{v}_{I_t}^{(t)})$, which subsequently leads to

$$\mathbf{A}_{I_t}^T \mathbf{A}_{I_{t-1}}(\mathbf{w}_{t-1} - \mathbf{v}_{I_{t-1}}^{(t-1)}) \in \partial\phi_t(\mathbf{v}_{I_t}^{(t)}) \tag{196}$$

By lemma 9, we obtain that

$$\begin{aligned}
\frac{1}{\sqrt{m}}\|\mathbf{w}_t - \mathbf{v}_{I_t}^{(t)}\|_2 &\leqslant \frac{1}{(1-\delta_k)\sqrt{m}}\|\mathbf{A}_{I_t}^T \mathbf{A}_{I_{t-1}}(\mathbf{w}_{t-1} - \mathbf{v}_{I_{t-1}}^{(t-1)})\|_2 \\
&\leqslant \frac{\theta_k}{(1-\delta_k)\sqrt{m}}\|\mathbf{w}_{t-1} - \mathbf{v}_{I_{t-1}}^{(t-1)}\|_2 \\
&= \frac{\theta_k}{(1-\delta_k)\sqrt{m}}\|\mathbf{v}^{(t)} - \mathbf{v}^{(t-1)}\|_2 \\
&\leqslant \frac{\theta_k}{1-\delta_k}\frac{\mu L}{1-\delta_k}\left(\frac{\theta_k}{1-\delta_k}\right)^{t-1} = \frac{\mu L}{1-\delta_k}\left(\frac{\theta_k}{1-\delta_k}\right)^t
\end{aligned} \tag{197}$$

This proves (187) as $\|\mathbf{v}^{(t+1)} - \mathbf{v}^{(t)}\|_2 = \|\mathbf{w}_t - \mathbf{v}_{I_t}^{(t)}\|_2$. We also get that

$$\frac{1}{\sqrt{m}}\|\mathbf{A}_{I_{t+1}}^T \mathbf{A}_{I_t}(\mathbf{w}_t - \mathbf{v}_{I_t}^{(t)})\|_2 \leqslant \theta_k\|\mathbf{w}_t - \mathbf{v}_{I_t}^{(t)}\|_2 \leqslant \frac{\theta_k \mu L}{1 - \delta_k}\left(\frac{\theta_k}{1-\delta_k}\right)^t \tag{198}$$

Thus,

$$\begin{aligned}
\|\boldsymbol{\zeta}_{(I_t \cup I_{t+1})^c}^{(t+1)} - \boldsymbol{\zeta}_{(I_t \cup I_{t+1})^c}^{(t)}\|_\infty &= \|\mathbf{A}_{(I_t \cup I_{t+1})^c}^T \mathbf{A}_{I_t}(\mathbf{w}_t - \mathbf{v}_{I_t}^{(t)})\|_\infty \\
&\leqslant \min|\mathbf{A}_{I_{t+1}}^T \mathbf{A}_{I_t}(\mathbf{w}_t - \mathbf{v}_{I_t}^{(t)})| \leqslant \sqrt{\tfrac{1}{k}}\|\mathbf{A}_{I_{t+1}}^T \mathbf{A}_{I_t}(\mathbf{w}_t - \mathbf{v}_{I_t}^{(t)})\|_2 \\
&\leqslant \sqrt{\tfrac{m}{k}}\mu L\left(\tfrac{\theta_k}{1-\delta_k}\right)^{t+1}
\end{aligned} \tag{199}$$

which proves (189).

## Step 3:

It is now clear from (187) that if $\theta_k + \delta_k < 1$, the sequence $\mathbf{v}^t$ is absolutely convergent. Moreover, (188) and (189), together with the fact that

$$\frac{1}{\sqrt{m}}\|\boldsymbol{\zeta}_{I_{t+1}}^{t+1} - \boldsymbol{\zeta}_{I_{t+1}}^t\|_2 = \|\mathbf{A}_{I_{t+1}}^T \mathbf{A}_{I_t}(\mathbf{w}_t - \mathbf{v}_{I_t}^{(t)})\|_2 \leqslant \mu L\left(\frac{\theta_k}{1 - \delta_k}\right)^{t+1} \tag{200}$$

yield to

$$\begin{aligned}
\|\boldsymbol{\zeta}^{t+1} - \boldsymbol{\zeta}^t\|_2 &= \sqrt{\|\boldsymbol{\zeta}_{I_t}^{t+1} - \boldsymbol{\zeta}_{I_t}^t\|_2^2 + \|\boldsymbol{\zeta}_{I_{t+1}}^{t+1} - \boldsymbol{\zeta}_{I_{t+1}}^t\|_2^2 + \|\boldsymbol{\zeta}_{(I_t \cup I_{t+1})^c}^{t+1} - \boldsymbol{\zeta}_{(I_t \cup I_{t+1})^c}^t\|_2^2} \\
&= \sqrt{\|\boldsymbol{\zeta}_{I_t}^{t-1} - \boldsymbol{\zeta}_{I_t}^t\|_2^2 + \|\boldsymbol{\zeta}_{I_{t+1}}^{t+1} - \boldsymbol{\zeta}_{I_{t+1}}^t\|_2^2 + \|\boldsymbol{\zeta}_{(I_t \cup I_{t+1})^c}^{t+1} - \boldsymbol{\zeta}_{(I_t \cup I_{t+1})^c}^t\|_2^2} \\
&\leqslant \sqrt{m}\sqrt{\mu^2 L^2\left(\tfrac{\theta_k}{1-\delta_k}\right)^{2t} + \mu^2 L^2\left(\tfrac{\theta_k}{1-\delta_k}\right)^{2t+2} + \mu^2 L^2(\tfrac{m}{k} - 1)\left(\tfrac{\theta_k}{1-\delta_k}\right)^{2t+2}}
\end{aligned} \tag{201}$$

which shows that the sequence $\zeta^{(t)}$ is also absolutely convergent. Denote the limits for $\zeta^{(t)}$ and $\mathbf{v}^{(t)}$ by $\zeta^{(\infty)}$ and $\mathbf{v}^{(\infty)}$, respectively.

We have that

$$\frac{1}{\sqrt{m}}\|\mathbf{v}^{(0)} - \mathbf{v}^{(\infty)}\|_2 \leqslant \sum_{t=0}^{\infty} \|\mathbf{v}^{(t+1)} - \mathbf{v}^{(t)}\|_2 \leqslant \sum_{t=0}^{\infty} \frac{\mu L}{1-\delta_k} \left(\frac{\theta_k}{1-\delta_k}\right)^t$$
$$= \frac{\mu L}{1-\delta_k-\theta_k} \tag{202}$$

Now, we show that $\|\zeta^{(\infty)}\|_\infty$ is bounded. To see this, consider any index $i$ and denote by $t_1 < t_2 < \ldots$ the iterations $t$, where $i \in I_t$. For $i \notin I_0$ due to (188), we have that

$$\zeta_i^{(\infty)} - \zeta_i^{(0)} = \sum_{t=0}^{\infty} \zeta_i^{(t+1)} - \zeta_i^{(t)} = \sum_{t \neq t_r, t \neq t_r+1} \zeta_i^{(t+1)} - \zeta_i^{(t)} = \sum_{t|i\in(I_t \cup I_{t+1})^c} \zeta_i^{(t+1)} - \zeta_i^{(t)} \tag{203}$$

which leads to

$$|\zeta_i^{(\infty)}| \leqslant \zeta_i^{(0)} + \sum_{t|i\in(I_t \cup I_{t+1})^c} |\zeta_i^{(t+1)} - \zeta_i^{(t)}| \leqslant \mu L\sqrt{\frac{m}{k}} + \mu L \sum_{t=0}^{\infty} \left(\frac{\theta_k}{1-\delta_k}\right)^{t+1}$$
$$= \leqslant \mu L \left(\sqrt{\frac{m}{k}} + \frac{\theta_k}{1-\delta_k-\theta_k}\right) \tag{204}$$

Similarly, for any $i \in I_0$, we have that

$$\zeta_i^{(\infty)} - \zeta_i^{(1)} = \sum_{t \geqslant 1|i\in(I_t \cup I_{t+1})^c} \zeta_i^{(t+1)} - \zeta_i^{(t)} \tag{205}$$

Thus, noting that $\zeta_{I_0}^{(1)} = 0$, we get that

$$|\zeta_i^{(\infty)}| \leqslant \mu L \sum_{t=1}^{\infty} \left(\frac{\theta_k}{1-\delta_k}\right)^{t+1} = \frac{\mu L\theta_k^2}{(1-\theta_k-\delta_k)(1-\delta_k)} \tag{206}$$

Together, we get that

$$\|\zeta^{(\infty)}\|_\infty \leqslant \mu L \left(\sqrt{\frac{m}{k}} + \frac{\theta_k}{1-\delta_k-\theta_k}\right) \tag{207}$$

Note that as $\zeta^{(t)} \in \partial\phi(\mathbf{v}^{(t)})$, we obtain that $\zeta^{(\infty)} \in \partial\phi(\mathbf{v}^{(\infty)})$. $\qquad\square$

The second claim in part 1 can be easily proved by a similar approach as in the previous theorems: Notice that for any sufficiently small value of $\delta > 0$,

$$\frac{\Phi_\lambda - \Phi_{\lambda-\delta}}{\delta} \leqslant \frac{\|\hat{\mathbf{x}}\|_1}{n} \leqslant \frac{\Phi_{\lambda+\delta} - \Phi_\lambda}{\delta} \tag{208}$$

This shows that

$$\frac{\|\hat{\mathbf{x}}\|_1}{n} \to_p \frac{\partial\Phi_\lambda}{\partial\lambda} = M_{\lambda|x|,|x|} \tag{209}$$

**Lemma 9.** *Consider the function $\rho(\mathbf{v}) = \frac{1}{2}\|\mathbf{h} + \mathbf{P}\mathbf{v}\|_2^2 + \lambda\|\mathbf{v} + \mathbf{x}\|_1 + \mathbf{p}^T\mathbf{v}$ and suppose that it is minimized at $\mathbf{v}^*$. Take an arbitrary point $\mathbf{v}$ and $\mathbf{q} \in \partial\rho(\mathbf{v})$. Then,*

$$\|\mathbf{v} - \mathbf{v}^*\|_2 \leqslant \frac{1}{\sigma_{min}^2(\mathbf{P})}\|\mathbf{q}\|_2 \tag{210}$$

*Proof.* Notice that the function $\rho$ can be written as $\rho(\mathbf{v}) = \frac{\alpha}{2}\|\mathbf{v}\|_2^2 + g(\mathbf{v})$, where $\alpha = \sigma_{min}(\mathbf{P})^2$ and $g(\mathbf{v})$ is convex. Now, we prove a more general result for any strongly convex function of the form $\rho(\mathbf{v}) = \frac{\alpha}{2}\|\mathbf{v}\|_2^2 + g(\mathbf{v})$, where $g$ is convex. For any point $\mathbf{v}$ any subgradient $\mathbf{p}$ of $f$, we have that $\mathbf{p} = \alpha\mathbf{v} + \mathbf{q}$, where $\mathbf{q}$ is a subgradient of $g$ at $\mathbf{v}$. Moreover at $\mathbf{v}^*$, $g$ has the subgradient $\mathbf{q}^* = -\alpha\mathbf{v}^*$. Since $g$ is convex, we have that

$$(\mathbf{q} - \mathbf{q}^*)^T(\mathbf{v} - \mathbf{v}^*) \geqslant 0 \tag{211}$$

which can also be written as

$$(\mathbf{p} + \alpha(\mathbf{v}^* - \mathbf{v}))^T(\mathbf{v} - \mathbf{v}^*) \geqslant 0 \Rightarrow \mathbf{p}^T(\mathbf{v} - \mathbf{v}^*) \geqslant \alpha\|\mathbf{v} - \mathbf{v}^*\|_2^2 \tag{212}$$

Using the Cauchy-Schwartz inequality, we obtain that $\|\mathbf{p}\|_2/\alpha \geqslant \|\mathbf{v} - \mathbf{v}^*\|_2$. Specializing this result for the given function and substituting $\alpha = \sigma_{min}(\mathbf{P})^2$, we obtain the desired result. $\qquad\square$

### 3.3.2  Part 2

**Step 1:** Denote by $\hat{\mathbf{x}}^{\lambda,\epsilon}$ the minimal solution of the optimization

$$\Phi_{\lambda,\epsilon} = \frac{1}{n} \min_{\mathbf{x}} \frac{1}{2}\|\mathbf{y} - \mathbf{A}\mathbf{x}\|_2^2 + \lambda\|\mathbf{x}\|_1 + \frac{\epsilon}{2}\|\mathbf{x}\|_2^2 \tag{213}$$

We later prove that under the given conditions, for each $\eta > 0$, there exist $\epsilon, \rho$ such that $0 < \epsilon < \eta$ and $|\rho| < \eta$. Moreover,

$$\Pr\left(\frac{\|\hat{\mathbf{x}}^{\lambda+\rho,\epsilon} - \hat{\mathbf{x}}^{\lambda,0}\|_2^2}{n} > \eta\right) \to 0 \tag{214}$$

Given this result, we may write that

$$\frac{g(\hat{\mathbf{x}}^{\lambda,0})}{n} - M_{\lambda|x|,g} = \frac{\sum_{i=1}^n g(\hat{x}_i^{\lambda,0})}{n} - M_{\lambda|x|,g} = \left(\frac{\sum_{i=1}^n g(\hat{x}_i^{\lambda,0})}{n} - \frac{\sum_{i=1}^n g(\hat{x}_i^{\lambda+\rho,\epsilon})}{n}\right) +$$

$$\left(\frac{\sum_{i=1}^n g(\hat{x}_i^{\lambda+\rho,\epsilon})}{n} - M_{(\lambda+\rho)|x|+\epsilon x^2/2,g}\right) +$$

$$\left(M_{(\lambda+\rho)|x|+\epsilon x^2/2,g} - M_{\lambda|x|,g}\right) \tag{215}$$

Define $\hat{\mathbf{x}}^{\lambda,0} - \mathbf{h} = \hat{\mathbf{x}}^{\lambda+\rho,\epsilon}$. Then from the Taylor expansion theorem, we have that

$$\frac{\sum_{i=1}^n g(\hat{x}_i^{\lambda,0})}{n} - \frac{\sum_{i=1}^n g(\hat{x}_i^{\lambda+\rho,\epsilon})}{n} = \frac{\sum_{i=1}^n g'(\hat{x}_i^{\lambda+\rho,\epsilon})h_i + g''(\eta_i)h_i^2/2}{n} \tag{216}$$

Using the Cauchy-Schwartz inequality and the fact that $g'' \leqslant C_1$ for some value of $C_1$, we get that

$$\left|\frac{\sum_{i=1}^n g(\hat{x}_i^{\lambda,0})}{n} - \frac{\sum_{i=1}^n g(\hat{x}_i^{\lambda+\rho,\epsilon})}{n}\right| \leqslant \sqrt{\frac{\sum_{i=1}^n (g')^2(\hat{x}_i^{\lambda+\rho,\epsilon})}{n}}\sqrt{\frac{\sum_{i=1}^n h_i^2}{n}} + \frac{C_1}{2}\frac{\sum_{i=1}^n h_i^2}{n} \tag{217}$$

Notice that since $g'' \leqslant C_1$, we have that $|g'(x)| \leqslant C_1|x| + C_2$. Then,

$$\frac{\sum_{i=1}^n (g')^2(\hat{x}_i^{\lambda+\rho,\epsilon})}{n} \leqslant 2C_2^2\frac{\sum_{i=1}^n (\hat{x}_i^{\lambda+\rho,\epsilon})^2}{n} + 2C_3^2 \tag{218}$$

From Theorem 1, the term $\dfrac{\sum_{i=1}^n (\hat{x}_i^{\lambda+\rho,\epsilon})^2}{n}$ converges in probability to a finite value. Hence, the exists a value $R > 0$, such that

$$\Pr\left(\frac{\sum_{i=1}^n (g')^2(\hat{x}_i^{\lambda+\rho,\epsilon})}{n} \geqslant R^2\right) \to 0 \tag{219}$$

Take an arbitrary value $\delta > 0$. Take $\eta_1 > 0$ such that $R\sqrt{\eta_1} + c_1\eta_1/2 < \delta/3$. Furthermore, it is easy to verify that, one can choose $\eta_2 > 0$ such that for any $0 < \epsilon < \eta_2, |\rho| < \eta_2$, we have that

$$\left|M_{(\lambda+\rho)|x|+\epsilon x^2/2,g} - M_{\lambda|x|,g}\right| < \frac{\delta}{3} \tag{220}$$

Next, take $\eta = \min(\eta_1, \eta_2)$. Assume the result in (214) with a proper choice of $\epsilon, \rho$ for the given value $\eta$. This leads to that with high probability

$$\frac{\sum_{i=1}^n h_i^2}{n} < \eta \leqslant \eta_1, \tag{221}$$

which further yields to

$$\left| \frac{\sum\limits_{i=1}^{n} g(\hat{x}_i^{\lambda,0})}{n} - \frac{\sum\limits_{i=1}^{n} g(\hat{x}_i^{\lambda+\rho,\epsilon})}{n} \right| \leqslant R\sqrt{\eta_1} + c_1\eta_1/2 < \delta/3 \tag{222}$$

Notice that from Theorem 1, we have that

$$\Pr\left( \left| \frac{\sum\limits_{i=1}^{n} g(\hat{x}_i^{\lambda+\rho,\epsilon})}{n} - M_{(\lambda+\rho)|x|+\epsilon x^2/2,g} \right| > \frac{\delta}{3} \right) \to 0 \tag{223}$$

From (215), (220), (222) and (223), we get that with high probability

$$\left| \frac{g(\hat{\mathbf{x}}^{\lambda,0})}{n} - M_{\lambda|x|,g} \right| \leqslant \delta \tag{224}$$

which leads to the desired result.

**Step 2:** It remains to show (214). First, observe that with high probability we have that

$$M_0 + \theta < \frac{l(1 - \delta_l(\mathbf{A}))}{2n} \tag{225}$$

where $\theta > 0$ is a fixed number and $l < n$ is a natural number, where $\delta_l < 1$. This shows that $(1-\delta_l) > 2(M_0+\theta)$ and $l/n > 2(M_0+\theta)$. Take $0 < \alpha < \min(4M_0, 2\theta)$. Define $K = M_0+\theta-\alpha/2$ and $k = \frac{l}{nK} - 1$. Notice that $K > M$ and

$$k = \frac{l}{n(M_0 + \theta - \alpha/2)} - 1 > \frac{l}{n(M_0 + \theta)} - 1 > 1 \tag{226}$$

Further,

$$K = M + \theta - \frac{\alpha}{2} \leqslant \frac{l(1 - \delta_l(\mathbf{A}))}{2n} \leqslant \frac{l}{n}\left[ \frac{1 - \alpha - \delta_l(\mathbf{A})}{2 - \alpha} + \alpha/2 \right] - \frac{\alpha}{2} \leqslant \frac{l}{n}\left[ \frac{1 - \alpha - \delta_l(\mathbf{A})}{2 - \alpha} \right] \tag{227}$$

which leads to

$$\alpha \leqslant \frac{k - 1 - (k + 1)\delta_l(\mathbf{A})}{k} \tag{228}$$

Denote $M^{\lambda,\epsilon} = M_{\lambda|x|+\frac{\epsilon}{2}x^2,x^2}$ and $N^{\lambda,\epsilon} = M_{\lambda|x|+\frac{\epsilon}{2}x^2,|x|}$. Take an arbitrary value $\delta > 0$. It is simple to see that there exist values $\rho, \epsilon$, such that $0 < \epsilon < \delta$, $|\rho| < \delta$ and $0 < N^{\lambda+\rho,\epsilon} - N^{\lambda,0} < \delta$. Then, take $\mu > 0$, such that

$$2\mu < N^{\lambda+\rho,\epsilon} - N^{\lambda,0}. \tag{229}$$

For the above values of $\epsilon$ and $\rho$, define $\mathbf{h} = \hat{\mathbf{x}}^{\lambda,0} - \hat{\mathbf{x}}^{\lambda+\rho,\epsilon}$. Denote the objective function in (213) by $\Phi_{\lambda,\epsilon}(\mathbf{x})$. Then we have

$$\begin{aligned}
\Phi_{\lambda+\rho,\epsilon}(\hat{\mathbf{x}}^{\lambda,0}) &= \Phi_{\lambda,0}(\hat{\mathbf{x}}^{\lambda,0}) + \frac{1}{n}\left( \frac{\epsilon}{2}\|\hat{\mathbf{x}}^{\lambda,0}\|_2^2 + \rho\|\hat{\mathbf{x}}^{\lambda,0}\|_1 \right) \leqslant \Phi_{\lambda,0}(\hat{\mathbf{x}}^{\lambda+\rho,\epsilon}) + \frac{1}{n}\left( \frac{\epsilon}{2}\|\hat{\mathbf{x}}^{\lambda,0}\|_2^2 + \rho\|\hat{\mathbf{x}}^{\lambda,0}\|_1 \right) \\
&= \Phi_{\lambda+\rho,\epsilon}(\hat{\mathbf{x}}^{\lambda+\rho,\epsilon}) + \frac{1}{n}\left( \frac{\epsilon}{2}\|\hat{\mathbf{x}}^{\lambda,0}\|_2^2 + \rho\|\hat{\mathbf{x}}^{\lambda,0}\|_1 - \frac{\epsilon}{2}\|\hat{\mathbf{x}}^{\lambda+\rho,\epsilon}\|_2^2 - \rho\|\hat{\mathbf{x}}^{\lambda+\rho,\epsilon}\|_1 \right) \\
&\leqslant \Phi_{\lambda+\rho,\epsilon} + \frac{\epsilon}{2}\frac{\|\mathbf{h}\|_2^2}{n} + \epsilon\frac{\|\mathbf{h}\|_2}{\sqrt{n}}\frac{\|\mathbf{x}^{\lambda+\rho,\epsilon}\|_2}{\sqrt{n}} + \frac{\rho}{n}\left( \|\hat{\mathbf{x}}^{\lambda,0}\|_1 - \|\hat{\mathbf{x}}^{\lambda+\rho,\epsilon}\|_1 \right)
\end{aligned} \tag{230}$$

Now, from Theorem 1 and the first part of Theorem 2 we have that

$$\frac{\|\hat{\mathbf{x}}^{\lambda+\rho,\epsilon}\|_2^2}{n} \to_p M^{\lambda+\rho,\epsilon}, \quad \frac{\|\hat{\mathbf{x}}^{\lambda+\rho,\epsilon}\|_1}{n} \to_p N^{\lambda+\rho,\epsilon}, \quad \frac{\|\hat{\mathbf{x}}^{\lambda,0}\|_1}{n} \to_p N^{\lambda,0} \tag{231}$$

Hence taking a constant value $M > \sqrt{M^{\lambda+\rho,\epsilon}}$, we obtain that

$$\Phi_{\lambda+\rho,\epsilon}(\hat{\mathbf{x}}^{\lambda,0}) \leqslant \Phi_{\lambda+\rho,\epsilon} + \frac{\epsilon}{2}\frac{\|\mathbf{h}\|_2^2}{n} + M\epsilon\frac{\|\mathbf{h}\|_2}{\sqrt{n}} + \rho\delta \tag{232}$$

Define the following index sets

$$S = \{k \mid |\hat{x}_k^{\lambda+\rho,\epsilon}| \geqslant \mu\} \quad L = \{k \mid 0 < |\hat{x}_k^{\lambda+\rho,\epsilon}| < \mu\} \tag{233}$$

Define $K_\mu^{\lambda,\epsilon} = M_{\lambda|x|+\epsilon x^2/2, \chi_{\mathbb{R}\setminus(-\mu,\ \mu)}}$. Notice that by Theorem 1, we have that

$$\frac{|S|}{n} \to_p K_\mu^{\lambda+\rho,\epsilon} \tag{234}$$

On the other hand

$$\lim_{(\mu,\rho,\epsilon)\to 0} K_\mu^{\lambda+\rho,\epsilon} = M_0 \tag{235}$$

Hence, for small enough values of $\delta$, we have that $K_\mu^{\lambda+\rho,\epsilon} < K$, which subsequently yields to the fact that with high probability

$$\frac{|S|}{n} < K \tag{236}$$

From (231), we know that with high probability,

$$\frac{\|\hat{\mathbf{x}}^{\lambda+\rho,\epsilon}\|_1}{n} - \frac{\|\hat{\mathbf{x}}^{\lambda,0}\|_1}{n} > 2\mu \tag{237}$$

Which can also be written as,

$$\frac{\|\hat{\mathbf{x}}_S^{\lambda+\rho,\epsilon}\|_1}{n} + \frac{\|\hat{\mathbf{x}}_T^{\lambda+\rho,\epsilon}\|_1}{n} > \frac{\|\hat{\mathbf{x}}_S^{\lambda+\rho,\epsilon}+\mathbf{h}_S\|_1}{n} + \frac{\|\hat{\mathbf{x}}_T^{\lambda+\rho,\epsilon}+\mathbf{h}_T\|_1}{n} + \frac{\|\mathbf{h}_{(S\cup T)^c}\|_1}{n} + 2\mu$$

$$\geqslant \frac{\|\hat{\mathbf{x}}_S^{\lambda+\rho,\epsilon}\|_1-\|\mathbf{h}_S\|_1}{n} + \frac{\|\mathbf{h}_T\|_1-\|\hat{\mathbf{x}}_T^{\lambda+\rho,\epsilon}\|_1}{n} + \frac{\|\mathbf{h}_{(S\cup T)^c}\|_1}{n} + 2\mu \tag{238}$$

Notice that by definition $\|\hat{\mathbf{x}}_T^{\lambda+\rho,\epsilon}\|_1 \leqslant \mu$. Hence, we obtain that with high probability

$$\|\mathbf{h}_S\|_1 \geqslant \|\mathbf{h}_{S^c}\|_1 \tag{239}$$

Now, define $\mathbf{z} = \mathbf{y} - \mathbf{A}\hat{\mathbf{x}}^{(\lambda+\rho,\epsilon)}$. Decompose with the following procedure the vector $\mathbf{h}_{S^c}$ into the blocks $T_1, T_2, \ldots$: $\mathbf{h}_{T_1}$ is the $k|S|$ elements of $\mathbf{h}_{S^c}$ with the largest absolute value. $\mathbf{h}_{T_2}$ is the $k|S|$ elements of the remaining elements (i.e., the ones in $\mathbf{h}_{S^c \setminus T_1}$) with the largest absolute, and so on. Define $U = S \cup T_1$. We have that

$$n\Phi_{\lambda+\rho,\epsilon}(\hat{\mathbf{x}}^{\lambda,0}) = \frac{1}{2}\|\mathbf{z} - \mathbf{A}\mathbf{h}\|_2^2 + (\lambda+\rho)\|\mathbf{x}^{\lambda+\rho,\epsilon} + \mathbf{h}\|_1 + \frac{\epsilon}{2}\|\mathbf{x}^{\lambda+\rho,\epsilon} + \mathbf{h}\|_2^2 \tag{240}$$

Notice that $\hat{\mathbf{x}}^{\lambda,0} = \hat{\mathbf{x}}^{\lambda+\rho,\epsilon} + \mathbf{h}$ is the minimal point of the function $\Phi_{\lambda,0}(\mathbf{x})$. Hence,

$$\mathbf{A}^T(\mathbf{z} - \mathbf{A}\mathbf{h}) = \mathbf{A}^T(\mathbf{y} - \mathbf{A}\hat{\mathbf{x}}^{\lambda,0}) \in \lambda\partial\|\hat{\mathbf{x}}^{\lambda,0}\|_1 \tag{241}$$

Hence,

$$\|\mathbf{A}_{U^c}^T(\mathbf{z} - \mathbf{A}\mathbf{h})\|_\infty \leqslant \lambda \Rightarrow -\mathbf{h}_{U^c}^T\mathbf{A}_{U^c}^T(\mathbf{z} - \mathbf{A}\mathbf{h}) \geqslant -\lambda\|\mathbf{h}_{U^c}\|_1 \tag{242}$$

which leads to

$$-\mathbf{h}_{U^c}^T\mathbf{A}_{U^c}^T(\mathbf{z} - \mathbf{A}_U\mathbf{h}_U) \geqslant -\lambda\|\mathbf{h}_{U^c}\|_1 - \|\mathbf{A}_{U^c}\mathbf{h}_{U^c}\|_2^2 \tag{243}$$

Finally, we get that

$$\frac{1}{2}\|\mathbf{z}-\mathbf{A}\mathbf{h}\|_2^2 = \frac{1}{2}\|\mathbf{z}-\mathbf{A}_U\mathbf{h}_U\|_2^2 - \mathbf{h}_c^T\mathbf{A}_{U^c}^T(\mathbf{z}-\mathbf{A}_U\mathbf{h}_U) + \frac{1}{2}\|\mathbf{A}_{U^c}\mathbf{h}_{U^c}\|_2^2 \geqslant \frac{1}{2}\|\mathbf{z}-\mathbf{A}_U\mathbf{h}_U\|_2^2 - \lambda\|\mathbf{h}_{U^c}\|_1 - \frac{1}{2}\|\mathbf{A}_{U^c}\mathbf{h}_{U^c}\|_2^2 \tag{244}$$

Hence, we have

$$n\Phi_{\lambda+\rho,\epsilon}(\hat{\mathbf{x}}^{\lambda,0}) \geqslant \frac{1}{2}\|\mathbf{z} - \mathbf{A}_U\mathbf{h}_U\|_2^2 - \lambda\|\mathbf{h}_{U^c}\|_1 - \frac{1}{2}\|\mathbf{A}_{U^c}\mathbf{h}_{U^c}\|_2^2 + (\lambda+\rho)\|\mathbf{x}_U^{\lambda+\rho,\epsilon} + \mathbf{h}_U\|_1 + (\lambda+\rho)\|\mathbf{x}_{U^c}^{\lambda+\rho,\epsilon} + \mathbf{h}_{U^c}\|_1$$

$$+ \frac{\epsilon}{2}\|\mathbf{x}_U^{\lambda+\rho,\epsilon} + \mathbf{h}_U\|_2^2 + \frac{\epsilon}{2}\|\mathbf{x}_{U^c}^{\lambda+\rho,\epsilon} + \mathbf{h}_{U^c}\|_2^2 \tag{245}$$

Notice that $\mathbf{w} = 0$ is the minimum point of the function

$$\frac{1}{2}\|\mathbf{z} - \mathbf{A}_U\mathbf{w}\|_2^2 + (\lambda+\rho)\|\mathbf{x}_U^{\lambda+\rho,\epsilon} + \mathbf{w}\|_1 + \frac{\epsilon}{2}\|\mathbf{x}_U^{\lambda+\rho,\epsilon} + \mathbf{w}\|_2^2 \tag{246}$$

Hence, from lemma 10, we get that

$$\frac{1}{2}\|\mathbf{z} - \mathbf{A}_U\mathbf{h}_U\|_2^2 + (\lambda+\rho)\|\mathbf{x}_U^{\lambda+\rho,\epsilon} + \mathbf{h}_U\|_1 + \frac{\epsilon}{2}\|\mathbf{x}_U^{\lambda+\rho,\epsilon} + \mathbf{h}_U\|_2^2$$

$$\geqslant \frac{\sigma_{min}^2(\mathbf{A}_U)}{2}\|\mathbf{h}_U\|_2^2 + \frac{1}{2}\|\mathbf{z}\|_2^2 + (\lambda+\rho)\|\mathbf{x}_U^{\lambda+\rho,\epsilon}\|_1 + \frac{\epsilon}{2}\|\mathbf{x}_U^{\lambda+\rho,\epsilon}\|_2^2 \tag{247}$$

Substituting this result in (245) gives that

$$
n\Phi_{\lambda+\rho,\epsilon}(\hat{\mathbf{x}}^{\lambda,0}) - n\Phi_{\lambda+\rho,\epsilon}(\hat{\mathbf{x}}^{\lambda+\rho,\epsilon})
$$

$$
\geqslant \frac{\sigma_{min}^2(\mathbf{A}_U)}{2}\|\mathbf{h}_U\|_2^2 - \lambda\|\mathbf{h}_{U^c}\|_1 - \tfrac{1}{2}\|\mathbf{A}_{U^c}\mathbf{h}_{U^c}\|_2^2 + (\lambda+\rho)\|\mathbf{x}_{U^c}^{\lambda+\rho,\epsilon} + \mathbf{h}_{U^c}\|_1
$$
$$
- (\lambda+\rho)\|\mathbf{x}_{U^c}^{\lambda+\rho,\epsilon}\|_1 - \tfrac{\epsilon}{2}\|\mathbf{x}_{U^c}^{\lambda+\rho,\epsilon}\|_2^2 + \tfrac{\epsilon}{2}\|\mathbf{x}_{U^c}^{\lambda+\rho,\epsilon} + \mathbf{h}_{U^c}\|_2^2
$$
$$
\geqslant \frac{\sigma_{min}^2(\mathbf{A}_U)}{2}\|\mathbf{h}_U\|_2^2 + \rho\|\mathbf{h}_{U^c}\|_1 - \tfrac{1}{2}\|\mathbf{A}_{U^c}\mathbf{h}_{U^c}\|_2^2 - 2(\lambda+\rho)\|\mathbf{x}_{U^c}^{\lambda+\rho,\epsilon}\|_1 - 2\|\mathbf{x}_{U^c}^{\lambda+\rho,\epsilon}\|_2\|\mathbf{h}_{U^c}\|_2
$$
$$
\geqslant \frac{\sigma_{min}^2(\mathbf{A}_U)}{2}\|\mathbf{h}_U\|_2^2 - \delta\sqrt{n}\|\mathbf{h}_U\|_2 - \tfrac{1}{2}\|\mathbf{A}_{U^c}\mathbf{h}_{U^c}\|_2^2 - 2(\lambda+\rho)n\mu - 2\sqrt{n}\mu\|\mathbf{h}_{U^c}\|_2 \quad (248)
$$

where we used the fact that

$$
\rho\|\mathbf{h}_{U^c}\|_1 \geqslant -\delta\|\mathbf{h}_{U^c}\|_1 \geqslant -\delta\|\mathbf{h}_U\|_1 \geqslant -\delta\sqrt{n}\|\mathbf{h}_U\|_2 \tag{249}
$$

In [4, Equation (11)] it is proved that

$$
\|\mathbf{h}_{U^c}\|_2^2 \leqslant \frac{|S|}{|T|}\|\mathbf{h}_U\|_2^2 = \frac{1}{k}\|\mathbf{h}_U\|_2^2 \tag{250}
$$

Also, in [Candes eq (12)] it is shown that

$$
\|\mathbf{A}_{U^c}\mathbf{h}_{U^c}\|_2 \leqslant \sqrt{1 + \delta_{k|S|}(\mathbf{A})}\sqrt{\frac{|S|}{|T|}}\|\mathbf{h}_U\|_2 = \sqrt{\frac{1 + \delta_{k|S|}(\mathbf{A})}{k}}\|\mathbf{h}_U\|_2 \tag{251}
$$

Hence,

$$
n\Phi_{\lambda+\rho,\epsilon}(\hat{\mathbf{x}}^{\lambda,0}) - n\Phi_{\lambda+\rho,\epsilon}(\hat{\mathbf{x}}^{\lambda+\rho,\epsilon})
$$
$$
\geqslant \left( \frac{1 - \delta_{(1+k)|S|}(\mathbf{A}) - \frac{1+\delta_{k|S|}(\mathbf{A})}{k}}{2} \right)\|\mathbf{h}_U\|_2^2 - (1 + \tfrac{1}{\sqrt{k}})\delta\sqrt{n}\|\mathbf{h}_U\|_2 - (\lambda+\delta)n\delta \tag{252}
$$

Notice that $|S| < Kn$. Hence according to (228),

$$
\alpha_1 = 1 - \delta_{(1+k)|S|}(\mathbf{A}) - \frac{1 + \delta_{k|S|}(\mathbf{A})}{k} \geqslant 1 - \delta_{n(1+k)K}(\mathbf{A}) - \frac{1 + \delta_{nkK}(\mathbf{A})}{k} \geqslant 1 - \delta_l(\mathbf{A}) - \frac{1 + \delta_l(\mathbf{A})}{k} \geqslant \alpha \tag{253}
$$

which gives that

$$
n\Phi_{\lambda+\rho,\epsilon}(\hat{\mathbf{x}}^{\lambda,0}) - n\Phi_{\lambda+\rho,\epsilon}(\hat{\mathbf{x}}^{\lambda+\rho,\epsilon})
$$
$$
\geqslant \tfrac{\alpha}{2}\|\mathbf{h}_U\|_2^2 - (1 + \tfrac{1}{\sqrt{k}})\delta\sqrt{n}\|\mathbf{h}_U\|_2 - (\lambda+\delta)n\delta \tag{254}
$$

Combining (232) and (232), we get that

$$
\frac{\alpha}{2}\|\mathbf{h}_U\|_2^2 - (1 + \frac{1}{\sqrt{k}})\delta\sqrt{n}\|\mathbf{h}_U\|_2 - (\lambda+\delta)n\delta \leqslant \frac{\delta}{2}\|\mathbf{h}\|_2^2 + M\delta\sqrt{n}\|\mathbf{h}\|_2 + n\delta^2 \tag{255}
$$

Notice that

$$
\|\mathbf{h}\|_2^2 \leqslant (1 + \frac{1}{k})\|\mathbf{h}_U\|_2^2 \tag{256}
$$

Then, we get that

$$
\frac{\alpha}{2(1 + \frac{1}{k})}\|\mathbf{h}\|_2^2 - \frac{1 + \frac{1}{\sqrt{k}}}{\sqrt{1 + \frac{1}{k}}}\delta\sqrt{n}\|\mathbf{h}\|_2 - (\lambda+\delta)n\delta \leqslant \frac{\delta}{2}\|\mathbf{h}\|_2^2 + M\delta\sqrt{n}\|\mathbf{h}\|_2 + n\delta^2 \tag{257}
$$

Since $k > 1$, it is simple to see for any $\eta > 0$ that the value $\delta$ can be made sufficiently small such that (257) implies (214).

**Lemma 10.** *Consider the function* $\rho(\mathbf{v}) = \frac{1}{2}\|\mathbf{h} + \mathbf{P}\mathbf{v}\|_2^2 + \lambda\|\mathbf{v} + \mathbf{x}\|_1 + \frac{\epsilon}{2}\|\mathbf{v}\|_2^2$ *and suppose that it is minimized at* $\mathbf{v}^*$. *Take an arbitrary point* $\mathbf{v}$. *Then,*

$$
\rho(\mathbf{v}) - \rho(\mathbf{v}^*) \geqslant \frac{\sigma_{min}^2(\mathbf{P})}{2}\|\mathbf{v} - \mathbf{v}^*\|_2^2 \tag{258}
$$

*Proof.* Define $\mathbf{w} = \frac{\mathbf{v}-\mathbf{v}^*}{\|\mathbf{v}-\mathbf{v}^*\|_2}$ and $f(v) = \rho(\mathbf{v}^* + v\mathbf{w})$. Notice that $\rho(\mathbf{v}) = f(\|\mathbf{v} - \mathbf{v}^*\|_2)$ and $f$ is minimized at 0. Moreover, direct calculations shows that $f$ can be written as $f = \frac{1}{2}\alpha v^2 + g(v)$, where $g$ is convex and $\alpha = \|\mathbf{Pw}\|_2^2 + \epsilon/2 \geqslant \sigma_{min}(\mathbf{P})^2$. Then, Lemma 11 leads to

$$\rho(\mathbf{v}) - \rho(\mathbf{v}^*) = f(\|\mathbf{v} - \mathbf{v}^*\|_2) - f(0) \geqslant \frac{\alpha}{2}\|\mathbf{v} - \mathbf{v}^*\|_2^2 \geqslant \frac{\sigma_{min}^2(\mathbf{P})}{2}\|\mathbf{v} - \mathbf{v}^*\|_2^2 \tag{259}$$

$\square$

**Lemma 11.** *Suppose that $g(v)$ is a convex function on $\mathbb{R}$ and $v^*$ is a minimum point of the function $f(v) = \frac{\alpha}{2}v^2 + g(v)$. Then, for any $v \in \mathbb{R}$,*

$$f(v) - f(v^*) \geqslant \frac{\alpha}{2}(v - v^*)^2 \tag{260}$$

*Proof.* From the optimality of $v^*$, we have that $-\alpha v^* \in \partial g(v^*)$. Hence,

$$g(v) \geqslant g(v^*) - \alpha v^*(v - v^*) \tag{261}$$

Hence,

$$f(v) - f(v^*) = \alpha v^*(v - v^*) + \frac{\alpha}{2}(v - v^*)^2 + g(v) - g(v^*) \geqslant \frac{\alpha}{2}(v - v^*)^2 \tag{262}$$

$\square$

### 3.4   Proof of Theorem 2 in Paper

Let us first consider convexity over $p$: Since convexity is preserved by the linear action of expectation, we only require to show that $S_f\left(\frac{\beta}{p}, p\Gamma + X\right)$ is a convex function of $p$ for any realization of $\Gamma, X$ and $\beta$. We have that

$$S_f\left(\frac{\beta}{p}, p\Gamma + X\right) = \min_x \frac{\beta}{2p}(x - p\Gamma - X)^2 + f(x) \tag{263}$$

Now, notice that $\frac{\beta}{2p}(x - p\Gamma - X)^2$ is a jointly convex function of $x$ and $p$ (i.e., it is a convex function of the $2 \times 1$ vector $(x, p)$). To see this, notice that its epigraph

$$\left\{(x, p, A) \mid \frac{\beta}{2p}(x - p\Gamma - X)^2 < A, \; p > 0\right\} = \left\{(x, p, A) \mid \beta(x - p\Gamma - X)^2 - Ap < 0, p > 0\right\}$$

is a convex set (This is simply seen by introducing the linear transformation $a = (A - p)/2, b = (A + p)/2$ and $c = x - p\Gamma - X$, and checking that the condition $p > 0$ restricts the transformed set to the upper part of a circular cone, which is convex). As a result, the objective function $L(p, x)$ in (263) is jointly convex for $x$ and $p$. Take two values $p_1, p_2 > 0$ and their corresponding minimum solutions $x_1^*, x_2^*$ in (263). Also denote $S_f\left(\frac{\beta}{p}, p\Gamma + X\right)$ by $S(p)$ for simplicity. We have that

$$S(\theta p_1 + (1-\theta)p_2) = \min_x L(\theta p_1 + (1-\theta)p_2, x) \leqslant L(\theta p_1 + (1-\theta)p_2, \theta x_1^* + (1-\theta)x_2^*) \leqslant \theta S(p_1) + (1-\theta)S(p_2),$$

where $0 \leqslant \theta \leqslant 1$ is arbitrary. This shows that $S$ is convex and completes the proof.

Now, we consider concavity of $\psi(\beta)$: Notice that we may write

$$\psi(\beta) = \min_Y \mathcal{E}\left(\min_{p>0} \frac{p\beta(\gamma - 1)}{2} + \frac{\gamma\sigma^2\beta}{2p} - \frac{\gamma\beta^2}{2} + \frac{\beta}{2p}(Y - p\Gamma - X)^2 + f(Y)\right),$$

where $Y$ ranges over all real-valued random variables. Notice that the inner optimization (over $p$) is in the form $A(p) + \beta B(p) - \gamma\beta^2/2 + f(Y)$ where $A, B$ are convex functions of $p$. Hence, its optimal value is a concave function of $\beta$. Denoting this optimal value by $L_Y(\beta)$, we observe that $\psi = \min_Y \mathcal{E}(L_Y(\beta))$. Notice that the minimum of a family of concave functions is concave and $\mathcal{E}(L_Y(\beta))$ is a concave function of $\beta$. Hence, $\psi$ is concave.

## Footnotes

[1]This means than there exists a positive number $\epsilon$ such that $M_0 < M_{\mathrm{adm}} - \epsilon$ holds with high probability.