[Reviews · NeurIPS 2017]

Reviewer 1



The paper tackles the problem of penalized least squares when the dimension scales linearly with the sample size and the penalty separates the variables (e.g., LASSO). The authors prove asymptotic results for the solution and its difference with the true (yet unknown) vector. The main assumptions are on the design matrix and the unknown vector, which are both assumed to be random, both with iid entries. In other words, the authors study how input distributions are processed by penalized least squares by describing the output distribution (i.e., the distribution of the solution), when the input distribution has an iid structure. As one of their main tools, the authors prove that the initial high-dimensional optimization problem is tied to two low dimensional optimization problems. Major comments: - As a reader, I am very confused by the structure of the supplementary material. Some results are presented and labeled in the main text, but the labels (i.e., "Theorem 2") are used for other results in the Supplementary material, making it impossible to find the proofs of the results of the main text. The supplementary material is presented as a whole work, that could almost be submitted as is. However, its role should be to complete the main text with intermediate results and proofs. I believe it should be rewritten completely, with proofs that actually correspond to the results presented in the main text. If the authors wish to complete the main text and extend it with further results, they should rather consider to submit a longer version of their work to a conference. - I am not very familiar with this literature, but the assumption that the entries of the unknown vector x_0 are iid seems very artificial and not realistic to me: It gives symmetric and interchangeable roles to all the covariates in the regression. However, I think that on a theoretical level, the results are very interesting and already seem to require very technical proofs. I believe this fact should be emphasized in the main text (e.g., "true vectors with a different structure" are mentioned in Line 278 of the main text, but it is not clear what the authors have in mind and whether their results could be easily extended to the case where x_0 has independent entries that are not iid). Minor comments in the main text: - The way Equation (1) is written, the factor 1/n is confusing (it may be better to remove it, only for the reader's ease) - At first reading, it is not clear what the curly E is (e.g., in Equation (4)) - Figure 2: There are no vertical lines that depict optimal values of \lambda - Figure 3: Should it be \nu=2 or 3 ? (cf. Line 165) Minor comments in the supplementary material: - There is a confusion between f and f_n (I recommend that in the main text, Equations (1) and (3), f be replaced with f_n) - Line 97: Replace the second occurrence of 1/n f_n^(k) with f_n - Line 191, Equation (69): The 4 should be an exponent in "z4" I believe that the results presented in this work are worth acceptance at NIPS, but the paper - especially the supplementary material - needs to be rewritten.

Reviewer 2



The paper studies the universality of high-dimensional regularized least squares problems with respect to the distribution of the design matrix. The main result of the paper is that, under moment conditions on the entries of the design matrix A, the optimal value of the least squares problem, and characteristics of the optimal solution are independent of the details of the distribution of the A_ij’s. The proof uses the Lindeberg interpolation method, in a clever incremental way, to transform the problem to its Gaussian counterpart while preserving the characteristics of the optimization, then uses Gaussian comparison theorems (variants of Gordon’s escape through a mesh result) to analyze the Gaussian setting. The consequences are nice formulas predicting the exact values of the cost, the limiting empirical distribution of the entries of the solution and of the error vector. These formulas were previously known to be correct only in restricted settings. These exact asymptotic characterizations of random estimation problems have been studied by several authors and in different settings. There are similarities, at least in spirit, to the work of El Karoui [1], and Donoho-Montanari [2] on robust high-dimensional regression (a problem that could be seen as the dual of the one considered here), and to many papers that have appeared in the information-theory community on “single letter formulas” characterizing the MMSE of various estimation channels (linear regression, noisy rank-one matrix estimation, CDMA,…) See e.g., [3,4,5] and references therein. Although most of these efforts have mainly dealt with the Gaussian design setting, some have proved “channel universality”, which is essentially the focus of this paper. The authors should probably fill this bibliographic gap. Second, I would like to point out a different approach to the problem, inspired by rigorous work in statistical physics, and that will make the connection to the above mentioned work clearer: one could consider the log-sum-exp approximation to the optimization problem, with a “inverse temperature” parameter \beta. And one recovers the optimization in the limit \beta -> \infty. One could study the convergence of log-partition function obtained in this way, and obtain a formula for it, which in this setting would bare the name of “the replica-symmetric (RS) formula”. The “essential optimization” problem found by the authors is nothing else but the zero-temperature limit of this RS formula. (One would of course have to justify swapping the limits n \to \infty and \beta \to \infty.) This being said, the techniques used in this work seem to be of a different nature, and add to the repertoire of approaches to compute asymptotic values of these challenging problems. [1] Asymptotic behavior of unregularized and ridge-regularized high-dimensional robust regression estimators : rigorous results https://arxiv.org/abs/1311.2445 [2] High Dimensional Robust M-Estimation: Asymptotic Variance via Approximate Message Passing https://arxiv.org/abs/1310.7320 [3] The Mutual Information in Random Linear Estimation : https://arxiv.org/abs/1607.02335 [4] Mutual information for symmetric rank-one matrix estimation: A proof of the replica formula https://arxiv.org/abs/1606.04142 [5] Fundamental limits of symmetric low-rank matrix estimation : https://arxiv.org/abs/1611.03888

Reviewer 3



This work considers the asymptotic performance of regularized least squares solutions with random design matrices. Expressions for the asymptotic error of Lasso recently appeared in the literature work when the design matrix is i.i.d. Gaussian. Here the authors extend these and show universality results where the design matrix has i.i.d. non-Gaussian entries. The regularization function can also be a separable arbitrary strongly convex function, or the L1 norm. They define a two dimensional saddle point problem, whose solution can be used to find the distribution of the estimate and its error which also involves a proximal map. They also provide limited numerical result which exhibit the predicted universality of the Lasso error with Bernoulli and t-distributed design matrices. Please see below for additional comments. 1. The tractability of evaluating the expectation of 4 seems to be limited. Are there other regularization functions where a statement similar to section 3.1.1 is possible ? For instance does the analysis work with L2 regularization when n larger than m. 2. The numerical illustration of the universality phenomenon seems very limited. What are other distributions that obey the theorem statement ? (For instance Bernoulli with different parameters). 3. It is not clear how sparse random design matrices fit into the main conclusion. When the design is very sparse, its nullspace will inevitably contain sparse matrices and recovery is not possible. 4. The expectation notation seems to be non-standard. Is it defined over finite sample size, or asymptotically as n goes to infinity ? 5. Some typos line 104 'constant'. line 99 'and the error'